# META-FREE FEW-SHOT LEARNING VIA REPRESENTATION LEARNING WITH WEIGHT AVERAGING

## ABSTRACT

Recent studies on few-shot classification using transfer learning pose challenges to the effectiveness and efficiency of episodic meta-learning algorithms. Transfer learning approaches are a natural alternative, but they are restricted to few-shot classification. Moreover, little attention has been on the development of probabilistic models with well-calibrated uncertainty from few-shot samples, except for some Bayesian episodic learning algorithms. To tackle the aforementioned issues, we propose a new transfer learning method to obtain accurate and reliable models for few-shot regression and classification. The resulting method does not require episodic meta-learning and is called meta-free representation learning (MFRL). MFRL first finds low-rank representation generalizing well on meta-test tasks. Given the learned representation, probabilistic linear models are fine-tuned with few-shot samples to obtain models with well-calibrated uncertainty. The proposed method not only achieves the highest accuracy on a wide range of few-shot learning benchmark datasets but also correctly quantifies the prediction uncertainty. In addition, weight averaging and temperature scaling are effective in improving the accuracy and reliability of few-shot learning in existing meta-learning algorithms with a wide range of learning paradigms and model architectures.

## 1 INTRODUCTION

Currently, the vast majority of few-shot learning methods are within the general paradigm of meta-learning (a.k.a. learning to learn) (Bengio et al., 1991; Schmidhuber, 1987; Thrun & Pratt, 1998), which learns multiple tasks in an episodic manner to distill transferrable knowledge (Vinyals et al., 2016; Finn et al., 2017; Snell et al., 2017). Although many episodic meta-learning methods report state-of-the-art (SOTA) performance, recent studies show that simple transfer learning methods with fixed embeddings (Chen et al., 2019; Tian et al., 2020) can achieve similar or better performance in few-shot learning. It is found that the effectiveness of optimization-based meta-learning algorithms is due to reusing high-quality representation, instead of rapid learning of task-specific representation (Raghu et al., 2020). The quality of the presentation is not quantitatively defined, except for some empirical case studies (Goldblum et al., 2020). Recent machine learning theories (Saunshi et al., 2021) indicate that low-rank representation leads to better sample efficiency in learning a new task. However, those theoretical studies are within the paradigm of meta-learning and do not reveal how to obtain low-rank representation for few-shot learning outside the realm of meta-learning. This motivates us to investigate ways to improve the representation for adapting to new few-shot tasks in a meta-free manner by taking the advantage of simplicity and robustness in transfer learning.

In parallel, existing transfer learning methods also have limitations. That is, the existing transfer learning methods may not find representation generalizing well to unseen few-shot tasks (Chen et al., 2019; Dhillon et al., 2020) , compared with state-of-the-art meta-learning methods (Ye et al., 2020; Zhang et al., 2020). Although some transfer learning methods utilize knowledge distillation and self-supervised training to achieve strong performance in few-shot classification, they are restricted to few-shot classification problems (Mangla et al., 2020; Tian et al., 2020). To the best of our knowledge, no transfer learning method is developed to achieve similar performance to meta-learning in few-shot regression. As such, it is desirable to have a transfer learning method that finds high-quality representation generalizing well to unseen classification and regression problems.

The last limitation of the existing transfer learning methods in few-shot learning is the lack of uncertainty calibration. Uncertainty quantification is concerned with the quantification of how likely certain outcomes are. Despite a plethora of few-shot learning methods (in fact, machine learning in general) to improve the point estimation accuracy, few methods are developed to get probabilistic models with improved uncertainty calibration by integrating Bayesian learning into episodic meta-training (Grant et al., 2018; Finn et al., 2018; Yoon et al., 2018; Snell & Zemel, 2021). Few-shot learning models can be used in risk-averse applications such as medical diagnosis (Prabhu et al., 2019). The diagnosis decision is made on not only point estimation but also probabilities associated with the prediction. The risk of making wrong decisions is significant when using uncalibrated models (Begoli et al., 2019). Thus, the development of proper fine-tuning steps to achieve well-calibrated models is the key towards practical applications of transfer learning in few-shot learning.

In this paper, we develop a simple transfer learning method as our own baseline to allow easy regularization towards more generalizable representation and calibration of prediction uncertainty. The regularization in the proposed transfer learning method works for regression and classification problems so that we can handle both problems within a common architecture. The calibration procedure is easily integrated into the developed transfer learning method to obtain few-shot learning models with good uncertainty quantification. Therefore, the resulting method, called Meta-Free Representation Learning (MFRL), overcomes the aforementioned limitations in existing transfer learning methods for few-shot learning. Our empirical studies demonstrate that the relatively overlooked transfer learning method can achieve high accuracy and well-calibrated uncertainty in few-shot learning when it is combined with the proper regularization and calibration. Those two tools are also portable to meta-learning methods to improve accuracy and calibration, but the improvement is less significant compared with that of transfer learning.

We use stochastic weight averaging (SWA) (Izmailov et al., 2018), which is agnostic to loss function types, as implicit regularization to improve the generalization capability of the representation. We also shed light on that the effectiveness of SWA is due to its bias towards low-rank representation. To address the issue of uncertainty quantification, we fine-tune appropriate linear layers during the meta-test phase to get models with well-calibrated uncertainty. In MFRL, hierarchical Bayesian linear models are used to properly capture the uncertainty from very limited training samples in few-shot regression, whereas the softmax output is scaled with a temperature parameter to make the few-shot classification model well-calibrated. Our method is the first one to achieve well-calibrated few-shot models by only fine-tuning probabilistic linear models in the meta-test phase, without any learning mechanisms related to the meta-training or representation learning phase.

Our contributions in this work are summarized as follows:

- We propose a transfer learning method that can handle both few-shot regression *and* classification problems with performance exceeding SOTA.
- For the first time, we empirically find the implicit regularization of SWA towards low-rank representation, which is a useful property in transferring to few-shot tasks.
- The proposed method results in well-calibrated uncertainty in few-shot learning models while preserving SOTA accuracy.
- The implicit regularization of SWA and temperature scaling factor can be applied to existing meta-learning methods to improve their accuracy and reliability in few-shot learning.

## 2 RELATED WORK

**Episodic meta-learning** approaches can be categorized into metric-based and optimization-based methods. Metric-based methods project input data to feature vectors through nonlinear embeddings and compare their similarity to make the prediction. Examples of similarity metrics include the weighted $L1$ metric (Koch et al., 2015), cosine similarity (Qi et al., 2018; Vinyals et al., 2016), and Euclidean distance to class-mean representation (Snell et al., 2017). Instead of relying on predefined metrics, learnable similarity metrics are introduced to improve the few-shot classification performance (Oreshkin et al., 2018; Sung et al., 2018). Recent metric-based approaches focus on developing task-adaptive embeddings to improve few-shot classification accuracy. Those task-adaptive embeddings include attention mechanisms for feature transformation (Fei et al., 2021; Gidaris & Komodakis, 2018; Ye et al., 2020; Zhang et al., 2021), graph neural networks (Garcia & Estrach,

2018), implicit class representation (Ravichandran et al., 2019), and task-dependent conditioning (Oreshkin et al., 2018; Yoon et al., 2020; 2019). Although metric-based approaches achieve strong performance in few-shot classification, they cannot be directly applied to regression problems.

Optimization-based meta-learning approaches try to find transferrable knowledge and adapt to new tasks quickly. An elegant and powerful meta-learning approach, termed model-agnostic meta-learning (MAML), solves a bi-level optimization problem to find good initialization of model parameters (Finn et al., 2017). However, MAML has a variety of issues, such as sensitivity to neural network architectures, instability during training, arduous hyperparameter tuning, and high computational cost. On this basis, some follow-up methods have been developed to simplify, stabilize and improve the training process of MAML (Antoniou et al., 2018; Flennerhag et al., 2020; Lee & Choi, 2018; Nichol et al., 2018; Park & Oliva, 2019). In practice, it is very challenging to learn high-dimensional model parameters in a low-data regime. Latent embedding optimization (LEO) attempts to learn low-dimensional representation to generate high-dimensional model parameters (Rusu et al., 2019). Meanwhile, R2-D2 (Bertinetto et al., 2019) and MetaOptNet (Lee et al., 2019) reduce the dimensionality of trainable model parameters by freezing feature extraction layers during inner loop optimization. Note that the proposed method is fundamentally different from R2-D2 and MetaOptNet because our method requires neither episodic meta-learning nor bi-level optimization.

**Transfer learning** approaches first learn a feature extractor on all training data through standard supervised learning, and then fine-tune a linear predictor on top of the learned feature extractor in a new task (Chen et al., 2019). However, vanilla transfer learning methods for few-shot learning do not take extra steps to make the learned representation generalizing well to unseen meta-test tasks. Some approaches in this paradigm are developed to improve the quality of representation and boost the accuracy of few-shot classification, including cooperative ensembles (Dvornik et al., 2019), knowledge distillation (Tian et al., 2020), and auxiliary self-supervised learning (Mangla et al., 2020). Nevertheless, those transfer learning methods are restricted to few-shot classification. MFRL aims to find representation generalizing well from the perspective of low-rank representation learning, which is supported by recent theoretical studies (Saunshi et al., 2021). Furthermore, MFLR is the first transfer learning method that can handle both few-shot regression and classification problems and make predictions with well-calibrated uncertainty.

## 3 BACKGROUND

### 3.1 EPISODIC META-LEARNING

In episodic meta-learning, the meta-training data contains $\mathcal{T}$ episodes or tasks, where the $\tau^{th}$ episode consists of data $\mathcal{D}_\tau = \{(\mathbf{x}_{\tau,j}, \mathbf{y}_{\tau,j})\}_{j=1}^{N_\tau}$ with $N_\tau$ samples. Tasks and episodes are used interchangeably in the rest of the paper. Episodic meta-learning algorithms aim to find common model parameters $\theta$ which can be quickly adapted to task-specific parameters $\phi_\tau$ ($\tau = 1, ..., \mathcal{T}$). For example, MAML-type algorithms assume $\phi_\tau$ is one or a few gradient steps away from $\theta$ (Finn et al., 2017; 2018; Grant et al., 2018; Yoon et al., 2018), while other meta-learning approaches assume that $\phi_\tau$ and $\theta$ share the parameters in the feature extractor and only differ in the top layer (Bertinetto et al., 2019; Lee et al., 2019; Snell et al., 2017).

### 3.2 STOCHASTIC WEIGHT AVERAGING

The idea of stochastic weight averaging (SWA) along the trajectory of SGD goes back to Polyak–Ruppert averaging (Polyak & Juditsky, 1992). Theoretically, weight averaging results in faster convergence for linear models in supervised learning and reinforcement learning(Bach & Moulines, 2013; Lakshminarayanan & Szepesvari, 2018). In deep learning, we are more interested in tail stochastic weight averaging (Jain et al., 2018), which averages the weights after $T$ training epochs. The averaged model parameters $\theta_{\mathrm{SWA}}$ can be computed by running $s$ additional training epochs using SGD

$$\theta_{\mathrm{SWA}} = \frac{1}{s} \sum_{i=T+1}^{T+s} \theta_i, \tag{1}$$

where $\theta_i$ denotes the model parameters at the end of the $i$-th epoch. SWA has been applied to supervised learning of deep neural neural networks to achieve higher test accuracy (Izmailov et al., 2018).

## 4 METHODOLOGY

The proposed method is a two-step learning algorithm: meta-free representation learning followed by fine-tuning. We employ SWA to make the learned representation low-rank and better generalize to meta-test data. Given a meta-test task, a new top layer is fine-tuned with few-shot samples to obtain probabilistic models with well-calibrated uncertainty. Note that MFRL can be used for both regression and classification depending on the loss function. The pseudocode of MFRL is presented in Appendix A.1.

### 4.1 REPRESENTATION LEARNING

Common representation can be learned via maximization of the likelihood of all training data with respect to $\theta$ rather than following episodic meta-learning. To do so, we group the data $\mathcal{D}_\tau = \{(\mathbf{x}_{\tau,j}, \mathbf{y}_{\tau,j})\}_{j=1}^{N_\tau}$ from all meta-training tasks into a single dataset $\mathcal{D}_{\mathrm{tr}}$. Given aggregated training data $\mathcal{D}_{\mathrm{tr}} = \{\mathbf{X}, \mathbf{Y}\}$, representation can be learned by maximizing the likelihood $p(\mathcal{D}_{\mathrm{tr}} \mid \theta)$ with respect to $\theta$. Let $\theta = [\theta_f, \mathbf{W}]$, where $\theta_f$ represents parameters in the feature extractor and $\mathbf{W}$ denotes the parameters in the top linear layer. The feature extractor $h(\mathbf{x}) \in \mathbb{R}^p$ is a neural network parameterized by $\theta_f$ and outputs a feature vector of dimension $p$. The specific form of the loss function depends on whether the task is regression or classification and can be given as follows:

$$\mathcal{L}_{RP}(\theta) = -\log p(\mathcal{D}_{\mathrm{tr}} \mid \theta) = \begin{cases} \mathcal{L}_{MSE}(\theta), & \text{regression} \\ \mathcal{L}_{CE}(\theta), & \text{classification} \end{cases}$$

where

$$\mathcal{L}_{MSE}(\theta) = \frac{1}{2N'} \sum_{\tau=1}^{\mathcal{T}} \sum_{j=1}^{N_\tau} \left( y_{\tau,j} - \mathbf{w}_\tau^\top h(\mathbf{x}_{\tau,j}) \right)^2, \tag{2}$$

$$\mathcal{L}_{CE}(\theta) = -\sum_{j=1}^{N'} \sum_{c=1}^{\mathcal{C}} y_{j,c} \log \frac{\exp(\mathbf{w}_c^\top h(\mathbf{x}_j))}{\sum_{c'=1}^{\mathcal{C}} \exp(\mathbf{w}_{c'}^\top h(\mathbf{x}_j))} \tag{3}$$

For regression problems, the model learns $\mathcal{T}$ regression tasks ($\mathbf{W} = [\mathbf{w}_1, ..., \mathbf{w}_{\mathcal{T}}] \in \mathbb{R}^{(p+1) \times \mathcal{T}}$) simultaneously using the loss function $\mathcal{L}_{MSE}$ given in Eq. 2, whereas the model learns a $\mathcal{C}$-class classification model [1] ($\mathbf{W} = [\mathbf{w}_1, ..., \mathbf{w}_{\mathcal{C}}] \in \mathbb{R}^{(p+1) \times \mathcal{C}}$) for classification problems using the loss function $\mathcal{L}_{CE}$ in Eq. 3. The loss function - either Eq. 2 or 3 - can be minimized through standard stochastic gradient descent, where $N' = \sum_{\tau=1}^{\mathcal{T}} N_\tau$ is the total number of training samples.

**Post-processing via SWA** Minimizing the loss functions in Eq. 2 and 3 by SGD may not necessarily result in representation that generalizes well to few-shot learning tasks in the meta-test set. The last hidden layer of a modern deep neural network is high-dimensional and may contain spurious features that over-fit the meta-training data. Recent meta-learning theories indicate that better sample complexity in learning a new task can be achieved via low-rank representation, whose singular values decay faster (Saunshi et al., 2021). We aim to find low-rank representation $\mathbf{\Phi} = h(\mathbf{X})$ without episodic meta-learning, which is equivalent to finding the conjugate kernel $K^{\mathrm{C}} = \mathbf{\Phi}\mathbf{\Phi}^\top$ with fast decaying eigenvalues. To link the representation with the parameter space, we can linearize the neural network by the first-order Taylor expansion at $\theta_T$ and get the finite width neural tangent kernel (NTK) $K^{\mathrm{NTK}}(\mathbf{X}, \mathbf{X}) = \mathbf{J}(\mathbf{X})\mathbf{J}(\mathbf{X})^\top$, where $\mathbf{J}(\mathbf{X}) = \nabla_\theta f_\theta(\mathbf{X}) \in \mathbb{R}^{N' \times |\theta|}$ is the Jacobian matrix, and $K^{\mathrm{NTK}}$ is a composite kernel containing $K^{\mathrm{C}}$ (Fan & Wang, 2020). The distributions of eigenvalues for $K^{\mathrm{NTK}}$ and $K^{\mathrm{C}}$ are empirically similar. Analyzing $K^{\mathrm{NTK}}$ could shed light on the properties of $K^{\mathrm{C}}$. In parallel, $K^{\mathrm{NTK}}$ shares the same eigenvalues of the Gauss-Newton matrix $G = \frac{1}{N'} \mathbf{J}(\mathbf{X})^\top \mathbf{J}(\mathbf{X})$. For linearized networks with squared loss, the Gauss-Newton matrix $G$ well

---

[1] $\mathcal{C}$ is the total number of classes in $\mathcal{D}_{\mathrm{tr}}$. Learning a $\mathcal{C}$-class classification model solves all possible tasks in the meta-training dataset because each task $\mathcal{D}_\tau$ only contains a subset of $\mathcal{C}$ classes.

approximates the Hessian matrix $H$ when $y$ is well-described by $f_\theta(x)$ (Martens, 2020). This is the case when SGD converges to $\theta_T$ within a local minimum basin. A Hessian matrix with a lot of small eigenvalues corresponds to a flat minimum, where the loss function is less sensitive to the perturbation of model parameters (Keskar et al., 2017). It is known that averaging the weights after SGD convergence in a local minimum basin pushes $\theta_T$ towards the flat side of the loss valley (He et al., 2019). As a result, SWA could result in a faster decay of eigenvalues in the kernel matrix, and thus low-rank representation. Our conjecture about SWA as implicit regularization towards low-rank representation is empirically verified in Section 5.

## 4.2 FINE-TUNING

After representation learning is complete, $\mathbf{W}$ is discarded and $\theta_f$ is frozen in a new few-shot task. Given the learned representation, we train a new probabilistic top layer in a meta-test task using few-shot samples. The new top layer will be configured differently depending on whether the few-shot task is a regression or a classification problem.

**In a few-shot regression task**, we learn a new linear regression model $y = \mathbf{w}^\top h(\mathbf{x}) + \epsilon$ on a fixed feature extractor $h(\mathbf{x}) \in \mathbb{R}^p$ with few-shot training data $\mathcal{D} = \{(\mathbf{x}_i, y_i)\}_{i=1}^n$, where $\mathbf{w}$ denotes the model parameters and $\epsilon$ is Gaussian noise with zero mean and variance $\sigma^2$. To avoid interpolation on few-shot training data ($n \ll p$), a Gaussian prior $p(\mathbf{w} \mid \lambda) = \prod_{i=0}^p \mathcal{N}(w_i \mid 0, \lambda)$ is placed over $\mathbf{w}$, where $\lambda$ is the precision in the Gaussian prior. However, it is difficult to obtain an appropriate value for $\lambda$ in a few-shot regression task because no validation data is available in $\mathcal{D}$.

Hierarchical Bayesian linear models can be used to obtain optimal regularization strength and grounded uncertainty estimation using few-shot training data only. To complete the specification of the hierarchical Bayesian model, the hyperpriors on $\lambda$ and $\sigma^2$ are defined as $p(\lambda) = \text{Gamma}(\lambda \mid a, b)$ and $p(\sigma^{-2}) = \text{Gamma}(\sigma^{-2} \mid c, d)$, respectively. The hyper-priors become very flat and non-informative when $a$, $b$, $c$ and $d$ are set to very small values. The posterior over all latent variables given the data is $p(\mathbf{w}, \lambda, \sigma^2 \mid \mathbf{X}, \mathbf{y})$, where $\mathbf{X} = \{\mathbf{x}\}_{i=1}^n$ and $\mathbf{y} = \{y_i\}_{i=1}^n$. However, the posterior distribution $p(\mathbf{w}, \lambda, \sigma^2 \mid \mathbf{X}, \mathbf{y})$ is intractable. The iterative optimization based approximate inference (Tipping, 2001) is chosen because it is highly efficient. The point estimation for $\lambda$ and $\sigma^2$ is obtained by maximizing the marginal likelihood function $p(\mathbf{y} \mid \mathbf{X}, \lambda, \sigma^2)$. The posterior of model parameters $p(\mathbf{w} \mid \mathbf{X}, \mathbf{y}, \lambda, \sigma^2)$ is calculated using the estimated $\lambda$ and $\sigma^2$. Previous two steps are repeated alternately until convergence.

The predictive distribution for a new sample $\mathbf{x}_*$ is

$$p(y_* \mid \mathbf{x}_*, \mathbf{X}, \mathbf{y}, \lambda, \sigma^2) = \int p(y_* \mid \mathbf{x}_*, \mathbf{w}, \sigma^2) \, p(\mathbf{w} \mid \mathbf{X}, \mathbf{y}, \lambda, \sigma^2) \, d\mathbf{w}, \tag{4}$$

which can be computed analytically because both distributions on the right hand side of Eq. 4 are Gaussian. Consequently, hierarchical Bayesian linear models avoids over-fitting on few-shot training data and quantifies predictive uncertainty.

**In a few-shot classification task**, a new logistic regression model is learned with the post-processed representation. A typical $K$-way $n$-shot classification task $\mathcal{D} = \{(\mathbf{x}_i, y_i)\}_{i=1}^{nK}$ consists of $K$ classes (different from meta-training classes) and $n$ training samples per class. Minimizing an un-regularized cross-entropy loss results in a significantly over-confident classification model because the norm of logistic regression model parameters $\mathbf{W} \in \mathbb{R}^{(p+1) \times K}$ becomes very large when few-shot training samples can be perfectly separated in the setting $nK \ll p$. A weighted $L2$ regularization term is added to the cross-entropy loss to mitigate the issue

$$\mathcal{L}(\mathbf{W}) = -\sum_{i=1}^{nK} \sum_{c=1}^{K} y_{i,c} \log \frac{\exp(\mathbf{w}_c^\top h(\mathbf{x}_i))}{\sum_{c'=1}^{K} \exp(\mathbf{w}_{c'}^\top h(\mathbf{x}_i))} + \lambda \sum_{c=1}^{K} \mathbf{w}_c^\top \mathbf{w}_c, \tag{5}$$

where $\lambda$ is the regularization coefficient, which affects the prediction accuracy and uncertainty. It is difficult to select an appropriate value of $\lambda$ in each of the meta-test tasks due to the lack of validation data in $\mathcal{D}$. We instead treat $\lambda$ as a global hyper-parameter so that the value of $\lambda$ should be determined based on the accuracy on meta-validation data. Note that the selected $\lambda$ with high validation accuracy does not necessarily lead to well calibrated classification models. As such, we introduce the temperature scaling factor (Guo et al., 2017) as another global hyper-parameter to

scale the softmax output. Given a test sample $\mathbf{x}_*$, the predicted probability for class $c$ becomes

$$p_c = \frac{\exp(\mathbf{w}_c^\top h(\mathbf{x}_*)/T)}{\sum_{c'=1}^{K} \exp(\mathbf{w}_{c'}^\top h(\mathbf{x}_*)/T)}, \tag{6}$$

where $T$ is the temperature scaling factor. In practice, we select the L2 regularization coefficient $\lambda$ and the temperature scaling factor $T$ as follows. At first, we set $T$ to 1, and do grid search on the meta-validation data to find the $\lambda$ resulting in the highest meta-validation accuracy. However, fine-tuning $\lambda$ does not ensure good calibration. It is the temperature scaling factor that ensures the good uncertainty calibration. Similarly, we do grid search of $T$ on the meta-validation set, and choose the temperature scaling factor resulting in the lowest expected calibration error (Guo et al., 2017). Note that different values of $T$ do not affect the classification accuracy because temperature scaling is accuracy preserving.

## 5 EXPERIMENTS

We follow the standard setup in few-shot learning literature. The model is trained on a meta-training dataset and hyper-parameters are selected based on the performance on a meta-validation dataset. The final performance of the model is evaluated on a meta-test dataset. The proposed method is applied to few-shot regression and classification problems and compared against a wide range of alternative methods.

### 5.1 FEW-SHOT REGRESSION RESULTS

Sine waves (Finn et al., 2017) and head pose estimation (Patacchiola et al., 2020) datasets are used to evaluate the performance of MFRL in few-shot regression. We use the same backbones in literature (Patacchiola et al., 2020) to make fair comparisons. Details of the few-shot regression experiments can be found in Appendix A.2.

The results for few-shot regression are summarized in Table 1. In the sine wave few-shot regression, MFRL outperforms all meta-learning methods, demonstrating that high-quality representation can be learned in supervised learning, without episodic meta-learning. Although DKT with a spectral mixture (SM) kernel achieves high accuracy, the good performance should be attributed to the strong inductive bias to periodic functions in the SM kernel (Wilson & Adams, 2013). Additional results for MFRL with different activation functions are reported in Appendix A.3. In the head pose estimation experiment, MFRL also achieves the best accuracy. In both few-shot regression problems, SWA results in improved accuracy, suggesting that SWA can improve the quality of features and facilitate the learning of downstream tasks. In Fig. 1, uncertainty is correctly estimated by the hierarchical Bayesian linear model with learned features using just 10 training samples.

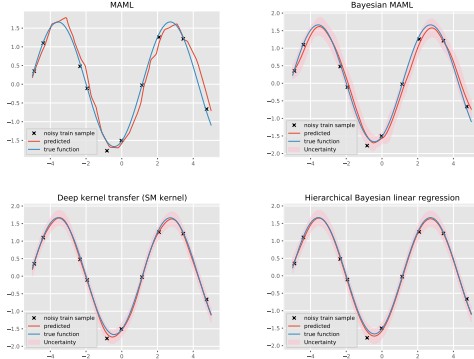

Figure 1: Sine wave regression and uncertainty quantification (10 training samples). The true and the estimated (by MFRL) standard deviation of data generation noise are 0.1, and 0.093.

| Sine wave (2-layer MLP) | MSE |
| --- | --- |
| MAML (Finn et al., 2017) | $0.67 \pm 0.06$ |
| Bayesian MAML (Yoon et al., 2018) | $0.54 \pm 0.05$ |
| ALPaCA (Harrison et al., 2018) | $0.14 \pm 0.09$ |
| R2D2 (Bertinetto et al., 2019) | $0.46 \pm$ NA |
| DKT+RBF (Patacchiola et al., 2020) | $1.38 \pm 0.03$ |
| DKT+Spectral (Patacchiola et al., 2020) | $0.08 \pm 0.06$ |
| MFRL (w.o. SWA) | $0.023 \pm 0.016$ |
| MFRL | $\mathbf{0.016 \pm 0.008}$ |
| Head pose (3-layer Conv Net) | MSE |
| MAML (Finn et al., 2017) | $0.21 \pm 0.01$ |
| Bayesian MAML (Yoon et al., 2018) | $0.18 \pm 0.01$ |
| DKT+Spectral (Patacchiola et al., 2020) | $0.10 \pm 0.01$ |
| MFRL (w.o. SWA) | $0.033 \pm 0.006$ |
| MFRL | $\mathbf{0.027 \pm 0.005}$ |

Table 1: 10-shot regression on sine waves and head pose estimation.

## 5.2 FEW-SHOT CLASSIFICATION RESULTS

We conduct few-shot classification experiments on four widely used few-shot image recognition benchmarks: miniImageNet (Ravi & Larochelle, 2017), tieredImageNet (Ren et al., 2018), CIFAR-FS (Bertinetto et al., 2019), and FC100 (Oreshkin et al., 2018). In addition, we test our approach on a cross-domain few-shot classification task from the miniImageNet to CUB. The experiment details about the few-shot classification datasets can be found in Appendix A.2. The proposed method is applied to three widely used network architectures: ResNet-12 (Lee et al., 2019; Ravichandran et al., 2019), wide ResNet (WRN-28-10) (Dhillon et al., 2020; Rusu et al., 2019), and a 4-layer convolutional neural network with 64 channels (Chen et al., 2019; Patacchiola et al., 2020) (in Appendix A.3).

Table 2: Few-shot classification results on miniImageNet and tieredImageNet.

| Method | Backbone | miniImageNet 5-way | | tieredImageNet 5-way | |
|---|---|---|---|---|---|
| | | 1-shot | 5-shot | 1-shot | 5-shot |
| Matching Net (Vinyals et al., 2016) | ResNet-12 | $63.08 \pm 0.80$ | $75.99 \pm 0.60$ | $68.50 \pm 0.92$ | $80.60 \pm 0.71$ |
| Proto Net (Snell et al., 2017) | ResNet-12 | $60.37 \pm 0.83$ | $78.02 \pm 0.57$ | $65.65 \pm 0.92$ | $83.40 \pm 0.65$ |
| Proto Net + SWA | ResNet-12 | $63.51 \pm 0.82$ | $81.98 \pm 0.58$ | $67.95 \pm 0.85$ | $84.76 \pm 0.66$ |
| MAML (Finn et al., 2017) | ResNet-12 | $56.58 \pm 1.84$ | $70.85 \pm 0.91$ | - | - |
| MAML + SWA | ResNet-12 | $58.21 \pm 1.86$ | $72.47 \pm 0.87$ | - | - |
| AdaResNet (Munkhdalai et al., 2018) | ResNet-12 | $56.88 \pm 0.62$ | $71.94 \pm 0.57$ | - | - |
| TADAM (Oreshkin et al., 2018) | ResNet-12 | $58.50 \pm 0.30$ | $76.70 \pm 0.30$ | - | - |
| Baseline++ (Chen et al., 2019) | ResNet-12 | $60.83 \pm 0.81$ | $77.81 \pm 0.76$ | $68.64 \pm 0.86$ | $80.47 \pm 0.67$ |
| Baseline++ + SWA | ResNet-12 | $65.72 \pm 0.80$ | $81.26 \pm 0.68$ | $70.01 \pm 0.82$ | $84.39 \pm 0.64$ |
| TapNet (Yoon et al., 2019) | ResNet-12 | $61.65 \pm 0.15$ | $76.36 \pm 0.10$ | $63.08 \pm 0.15$ | $80.26 \pm 0.12$ |
| MetaOptNet (Lee et al., 2019) | ResNet-12 | $62.64 \pm 0.61$ | $78.63 \pm 0.46$ | $65.99 \pm 0.72$ | $81.56 \pm 0.53$ |
| Ensemble (Dvornik et al., 2019) | ResNet-18 | $59.48 \pm 0.65$ | $75.62 \pm 0.42$ | - | - |
| DSN (Simon et al., 2020) | ResNet-12 | $62.64 \pm 0.66$ | $78.83 \pm 0.45$ | $66.22 \pm 0.75$ | $82.79 \pm 0.48$ |
| DKT (Patacchiola et al., 2020) | ResNet-12 | $61.29 \pm 0.57$ | $76.25 \pm 0.51$ | $67.21 \pm 0.52$ | $79.69 \pm 0.53$ |
| FEAT (Ye et al., 2020) | ResNet-12 | $66.78 \pm 0.20$ | $82.05 \pm 0.14$ | $70.80 \pm 0.23$ | $84.79 \pm 0.16$ |
| DeepEMD (Zhang et al., 2020) | ResNet-12 | $65.91 \pm 0.82$ | $82.41 \pm 0.56$ | $71.16 \pm 0.87$ | $86.03 \pm 0.58$ |
| Distill (Tian et al., 2020) | ResNet-12 | $64.82 \pm 0.60$ | $82.14 \pm 0.43$ | $71.52 \pm 0.69$ | $86.03 \pm 0.49$ |
| MFRL (w.o. SWA) | ResNet-12 | $62.27 \pm 0.86$ | $80.23 \pm 0.57$ | $70.03 \pm 0.77$ | $84.42 \pm 0.64$ |
| MFRL | ResNet-12 | $\mathbf{67.18 \pm 0.79}$ | $\mathbf{83.81 \pm 0.53}$ | $\mathbf{71.58 \pm 0.79}$ | $\mathbf{86.87 \pm 0.62}$ |
| LEO (Rusu et al., 2019) | WRN-28-10 | $61.76 \pm 0.08$ | $77.59 \pm 0.12$ | $66.33 \pm 0.03$ | $81.44 \pm 0.09$ |
| Fine-tune (Dhillon et al., 2020) | WRN-28-10 | $57.73 \pm 0.62$ | $78.17 \pm 0.49$ | $66.58 \pm 0.70$ | $85.55 \pm 0.48$ |
| Inductive SIB (Hu et al., 2020) | WRN-28-10 | $60.12 \pm 0.56$ | $78.17 \pm 0.35$ | $69.20 \pm 0.58$ | $84.96 \pm 0.36$ |
| MetaFun (Xu et al., 2020) | WRN-28-10 | $64.13 \pm 0.13$ | $80.82 \pm 0.17$ | $67.27 \pm 0.14$ | $83.28 \pm 0.12$ |
| MFRL (w.o. SWA) | WRN-28-10 | $61.83 \pm 0.82$ | $80.12 \pm 0.88$ | $69.89 \pm 0.79$ | $84.42 \pm 0.66$ |
| MFRL | WRN-28-10 | $\mathbf{66.42 \pm 0.80}$ | $\mathbf{82.26 \pm 0.61}$ | $\mathbf{71.47 \pm 0.84}$ | $\mathbf{86.34 \pm 0.65}$ |

Table 3: Few-shot classification results on CIFAR-FS and FC100.

| Method | Backbone | CIFAR-FS 5-way | | FC100 5-way | |
|---|---|---|---|---|---|
| | | 1-shot | 5-shot | 1-shot | 5-shot |
| Proto Net (Snell et al., 2017) | ResNet-12 | $72.2 \pm 0.7$ | $83.5 \pm 0.5$ | $41.5 \pm 0.7$ | $57.0 \pm 0.7$ |
| TADAM (Oreshkin et al., 2018) | ResNet-12 | - | - | $40.1 \pm 0.4$ | $56.1 \pm 0.4$ |
| Baseline++ (Chen et al., 2019) | ResNet-12 | $72.2 \pm 0.9$ | $84.2 \pm 0.6$ | $43.1 \pm 0.7$ | $55.7 \pm 0.7$ |
| MetaOptNet (Lee et al., 2019) | ResNet-12 | $72.8 \pm 0.7$ | $85.0 \pm 0.5$ | $41.1 \pm 0.6$ | $55.5 \pm 0.6$ |
| MTL (Sun et al., 2019) | ResNet-12 | - | - | $45.1 \pm 1.8$ | $57.6 \pm 0.9$ |
| Shot-free (Ravichandran et al., 2019) | ResNet-12 | $69.1 \pm \text{NA}$ | $84.7 \pm \text{NA}$ | - | - |
| TEAM (Qiao et al., 2019) | ResNet-12 | $70.4 \pm \text{NA}$ | $81.3 \pm \text{NA}$ | - | - |
| SIB (Hu et al., 2020) | ResNet-12 | $70.0 \pm 0.5$ | $83.5 \pm 0.4$ | - | - |
| DSN (Simon et al., 2020) | ResNet-12 | $72.3 \pm 0.8$ | $85.1 \pm 0.6$ | - | - |
| MABAS (Kim et al., 2020) | ResNet-12 | $73.5 \pm 0.9$ | $85.6 \pm 0.6$ | $42.3 \pm 0.7$ | $58.1 \pm 0.7$ |
| Distill (Tian et al., 2020) | ResNet-12 | $73.9 \pm 0.8$ | $86.9 \pm 0.5$ | $44.6 \pm 0.7$ | $60.9 \pm 0.6$ |
| MFRL (w.o. SWA) | ResNet-12 | $71.4 \pm 0.8$ | $86.1 \pm 0.5$ | $42.5 \pm 0.7$ | $59.1 \pm 0.7$ |
| MFRL | ResNet-12 | $\mathbf{74.0 \pm 0.8}$ | $\mathbf{87.4 \pm 0.6}$ | $\mathbf{45.3 \pm 0.8}$ | $\mathbf{61.1 \pm 0.7}$ |
| Fine-tune (Dhillon et al., 2020) | WRN-28-10 | $68.7 \pm 0.7$ | $86.1 \pm 0.6$ | $38.2 \pm 0.5$ | $57.2 \pm 0.6$ |
| MFRL (w.o. SWA) | WRN-28-10 | $71.7 \pm 0.9$ | $86.2 \pm 0.9$ | $41.5 \pm 0.7$ | $57.3 \pm 0.7$ |
| MFRL | WRN-28-10 | $\mathbf{76.7 \pm 0.9}$ | $\mathbf{88.6 \pm 0.5}$ | $\mathbf{45.1 \pm 0.8}$ | $\mathbf{61.0 \pm 0.8}$ |

The results of the proposed method and previous SOTA methods using similar backbones are reported in Table 2 and 3. The proposed method achieves the best performance in most of the experiments when compared with previous SOTA methods. Our method is closely related to Baseline++ (Chen et al., 2019) and fine-tuning on logits (Dhillon et al., 2020). Baseline++ normalizes both classification weights and features, while the proposed method only normalizes features. It allows our method to find a more accurate model in a more flexible hypothesis space, given high-quality representation. Compared with fine-tuning on logits, our method obtains better results by learning

a new logistic regression model on features, which store richer information about the data. Some approaches pretrain a $\mathcal{C}$-class classification model on all training data and then apply highly sophisticated meta-learning techniques to the pretrained model to achieve SOTA performance (Rusu et al., 2019; Sun et al., 2019). Our approach with SWA outperforms those pretrained-then-meta-learned models, which demonstrates that SWA obtains high-quality representation that generalizes well to unseen tasks. Compared with improving representation quality for few-shot classification via self-distillation (Tian et al., 2020), the computational cost of SWA is significantly smaller because it does not require training models from scratch multiple times. Moreover, SWA can be applied to find good representation for both few-shot regression and classification, while previous transfer learning approaches can only handle few-shot classification problems (Mangla et al., 2020; Tian et al., 2020).

MFRL is also applied to the cross-domain few-shot classification task as summarized in Table 4. MFRL outperforms other methods in this challenging task, indicating that the learned representation has strong generalization capability. We use the same hyperparameters (training epochs, learning rate, learning rate in SWA, SWA epoch, etc.) as in Table 2. The strong results indicate that MFRL is robust to hyperparameter choice. Surprisingly, meta-learning methods with adaptive embeddings do not outperform simple transfer learning methods like Baseline++ when the domain gap between base classes and novel classes is large. We notice that Tian et al. (2020) also reports similar results that transfer learning methods show superior performance on a large-scale cross-domain few-shot classification dataset. We still believe that adaptive embeddings should be helpful when the domain gap between base and novel classes is large. Nevertheless, how to properly train a model to obtain useful adaptive embeddings in novel tasks is an open question.

Table 4: Cross-domain few-shot classification results on miniImageNet to CUB.

| Method | Backbone | miniImageNet to CUB 5-way | |
| --- | --- | --- | --- |
| | | 1-shot | 5-shot |
| MAML (Finn et al., 2017) | WRN-28-10 | $39.06 \pm 0.47$ | $55.04 \pm 0.42$ |
| LEO (Rusu et al., 2019) | WRN-28-10 | $41.45 \pm 0.54$ | $56.66 \pm 0.48$ |
| MTL (Sun et al., 2019) | WRN-28-10 | $43.15 \pm 0.44$ | $56.89 \pm 0.41$ |
| Matching Net (Vinyals et al., 2016) | WRN-28-10 | $42.04 \pm 0.57$ | $53.08 \pm 0.45$ |
| SIB (Hu et al., 2020) | WRN-28-10 | $43.27 \pm 0.44$ | $59.94 \pm 0.42$ |
| Baseline (Chen et al., 2019) | WRN-28-10 | $42.89 \pm 0.41$ | $62.12 \pm 0.40$ |
| Baseline++ (Chen et al., 2019) | WRN-28-10 | $42.12 \pm 0.39$ | $60.21 \pm 0.39$ |
| MFRL (w.o. SWA) | WRN-28-10 | $43.68 \pm 0.47$ | $63.86 \pm 0.42$ |
| MFRL | WRN-28-10 | $\mathbf{46.98 \pm 0.51}$ | $\mathbf{66.92 \pm 0.42}$ |

## 5.3 Effective rank of the representation

The rank of representation defines the number of independent bases. For deep learning, noise in gradients and numerical imprecision can cause the resulting matrix to be full-rank. Therefore, simply counting the number of non-zero singular values may not be an effective way to measure the rank of the representation. To compare the effective ranks, we plot the normalized singular values of the representation of meta-test data in Fig. 2, where the representation with SWA has a faster decay in singular values, thus indicating the lower effective rank of the presentation with SWA. The results empirically verify our conjecture that SWA is an implicit regularizer towards low-rank representation.

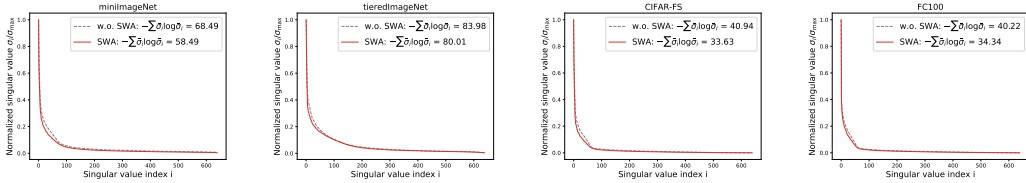

Figure 2: Normalized singular values for representation with and without SWA. The metric $-\sum \bar{\sigma}_i \log \bar{\sigma}_i$ is used to measure the effective rank of the representation, where $\bar{\sigma}_i = \sigma_i / \sigma_{\max}$. Faster decay in singular values indicates that fewer dimensions capture the most variation in all dimensions, thus lower effective rank.

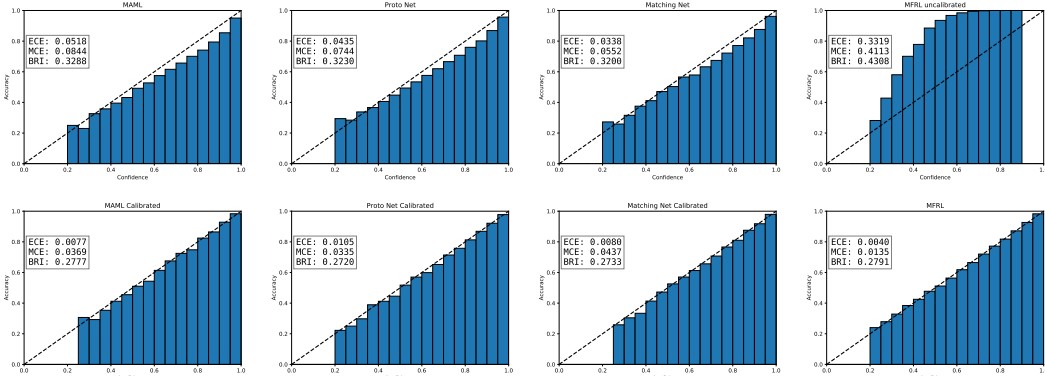

Figure 3: Study on the temperature scaling factor for 5-way 5-shot classification using ResNet-12 for the proposed MFRL and existing episodic meta learning methods. Uncalibrated models are in the first row, and calibrated models with the temperature scaling factor are in the second row.

## 5.4 FEW-SHOT CLASSIFICATION RELIABILITY

The proposed method not only achieves high accuracy in few-shot classification but also makes the classification uncertainty well-calibrated. A reliability diagram can be used to check model calibration visually, which plots an identity function between prediction accuracy and confidence when the model is perfectly calibrated (DeGroot & Fienberg, 1983). Fig. 3 shows the classification reliability diagrams along with widely used metrics for uncertainty calibration, including expected calibration error (ECE) (Guo et al., 2017), maximum calibration error (MCE) (Naeini et al., 2015), and Brier score (BRI) (Brier, 1950). ECE measures the average binned difference between confidence and accuracy, while MCE measures the maximum difference. BRI is the squared error between the predicted probabilities and one-hot labels. MAML is over-confident because tuning a deep neural network on few-shot data is prone to over-fitting. Meanwhile, Proto Net and Matching Net are better calibrated than MAML because they do not fine-tune the entire network during testing. Nevertheless, they are still slightly over-confident. The results indicate that MFRL with a global temperature scaling factor can learn well-calibrated models from very limited training samples.

## 5.5 APPLICATION IN META-LEARNING

Meanwhile, we also apply SWA to episodic meta-learning methods, such Proto Net, MAML and Matching Net, to improve their classification accuracy. The results in Table 5 indicate that SWA can improve the few-shot classification accuracy in both transfer learning and episodic meta-learning. SWA is orthogonal to the learning paradigm and model architecture. Thus, SWA can be applied to a wide range of few-shot learning methods to improve accuracy.

Table 5: Application of SWA on meta-learning methods for the miniImageNet dataset

| Method | Proto Net | | MAML | | Matching Net | |
|---|---|---|---|---|---|---|
| | 1-shot | 5-shot | 1-shot | 5-shot | 1-shot | 5-shot |
| w.o. SWA | 60.37 | 78.02 | 56.58 | 70.85 | 63.08 | 75.99 |
| SWA | 63.51 | 81.98 | 58.21 | 72.47 | 63.76 | 76.78 |

Furthermore, the temperature scaling factor can be applied to calibrate meta-learning methods, including MAML, Proto Net, and Matching Net. The reliability diagrams in Fig. 3 indicate that the temperature scaling factor not only calibrates classification uncertainty of transfer learning approaches, such as the proposed MFRL, but also makes the classification uncertainty well-calibrated in episodic meta-learning methods. Therefore, the temperature scaling factor can be applied to a wide range of few-shot classification methods to get well-calibrated uncertainty, while preserving the classification accuracy.

## 6 DISCUSSION

SWA has been applied to supervised learning of deep neural networks (Izmailov et al., 2018; Athiwaratkun et al., 2019) and its effectiveness was attributed to convergence to a solution on the flat side of an asymmetric loss valley (He et al., 2019). However, it does not explain the effectiveness of SWA in few-shot learning because the meta-training and meta-testing losses are not comparable after the top layer is retrained by the few-shot support data in a meta-test task. The effectiveness of SWA in few-shot learning must be related to the property of the representation. Although our results empirically demonstrate that SWA results in low-rank representation, further research about their connection is needed.

Explicit regularizers can also be used to obtain simple input-output functions in deep neural networks and low-rank representation, including L1 regularization, nuclear norm, spectral norm, and Frobenius norm (Bartlett et al., 2017; Neyshabur et al., 2018; Sanyal et al., 2020). However, some of those explicit regularizers are not compatible with standard SGD training or are computationally expensive. In addition, it is difficult to choose the appropriate strength of explicit regularization. Too strong explicit regularization can bias towards simple solutions that do not fit the data. In comparison, SWA is an implicit regularizer that is completely compatible with the standard SGD training without much extra computational cost. Thus, it can be easily combined with transfer learning and meta-learning to obtain more accurate few-shot learning models. In parallel, SWA is also robust to the choice of the hyperparameters - the learning rate and training epochs in the SWA stage (see details in Appendix A.4).

## 7 CONCLUSIONS

In this article, we propose MFRL to obtain accurate and reliable few-shot learning models. SWA is an implicit regularizer towards low-rank representation, which generalizes well to unseen meta-test tasks. The proposed method can be applied to both classification and regression tasks. Extensive experiments show that our method not only outperforms other SOTA methods on various datasets but also correctly quantifies the uncertainty in prediction.

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

## A  APPENDIX

### A.1  PSEUDO CODE FOR MFRL

---
**Algorithm 1** Meta-free representation learning for few-shot learning

---
Merge all training tasks $\mathcal{D}_{\mathrm{tr}} = \{\mathcal{D}_\tau\}_{\tau=1}^{\mathcal{T}}$
Initialize model parameters $\theta = [\theta_f, \mathbf{W}]$
Maximize the likelihood on all training data $p(\mathcal{D}_{\mathrm{tr}} \mid \theta)$ using SGD
    Minimize the squared loss for regression problems
    Minimize the cross-entropy loss for classification problems
Run SWA to obtain $\theta_{\mathrm{SWA}}$
Discard $\mathbf{W}$ and freeze $\theta_f$
Learn a new top layer using support data $\mathcal{D}$ in a test task:
    Learn a hierarchical Bayesian linear model for a regression task
    Learn a logistic regression model with the temperature scaling factor for a classification task

---

### A.2  EXPERIMENT DETAILS

**Sine waves** are generated by $y = A\sin(x - \varphi) + \epsilon$, where amplitude $A \in [0.1, 5.0]$, phase $\varphi \in [0, \pi]$ and $\epsilon$ is white noise with standard deviation of 0.1 (Finn et al., 2017). Each sine wave contains 200 samples by sampling $x$ uniformly from $[-5.0, 5.0]$. We generate 500 waves for training, validation and testing, respectively. All sine waves are different from each other. We use the same backbone network described in MAML (Finn et al., 2017): a two-layer MLP with 40 hidden units in each layer. We use the SGD optimizer with a learning rate of $10^{-3}$ over $8 \times 10^4$ training iterations and run SWA over $2 \times 10^4$ training iterations with a learning rate of 0.05.

**Head pose** regression data is derived from the Queen Mary University of London multi-view face dataset (Gong et al., 1996). It contains images from 37 people and 133 facial images per person. Facial images cover a view sphere of 90° in yaw and 120° in tilt. The dataset is divided into 3192 training samples (24 people), 1064 validation samples (8 people), and 665 test samples (5 people). We use the same feature extractor described in literature (Patacchiola et al., 2020): a three-layer convolutional neural network, each with 36 output channels, stride 2, and dilation 2. We train the model on the training people set for 300 epochs using the SGD optimizer with a learning rate of 0.01 and run 25 epochs of SWA with a learning rate of 0.01.

**miniImageNet** is a 100-class subset of the original ImageNet dataset (Deng et al., 2009) for few-shot learning (Vinyals et al., 2016). Each class contains 600 images in RGB format of the size 84 × 84. miniImageNet is split into 64 training classes, 16 validation classes, and 20 testing classes, following the widely used data splitting protocol (Ravi & Larochelle, 2017).

**tieredImageNet** is another subset of the ImageNet dataset for few-shot learning (Ren et al., 2018). It contains 608 classes grouped into 34 categories, which are split into 20 training categories (351 classes), 6 validation categories (97 classes), and 8 testing categories (160 classes). Compared with miniImageNet, training classes in tieredImageNet are sufficiently distinct from test classes, making few-shot classification more difficult.

**CIFAR-FS** is a derivative of the original CIFAR-100 dataset by randomly splitting 100 classes into 64, 16, and 20 classes for training, validation, and testing, respectively (Bertinetto et al., 2019).

**FC100** is another derivative of CIFAR-100 with minimized overlapped information between train classes and test classes by grouping the 100 classes into 20 superclasses (Oreshkin et al., 2018). They are further split into 60 training classes (12 superclasses), 20 validation classes (4 superclasses), and 20 test classes (4 superclasses).

**miniImageNet to CUB** is a cross-domain few-shot classification task, where the models are trained on miniImageNet and tested on CUB (Welinder et al., 2010). Cross-domain few-shot classification is more challenging due to the big domain gap between two datasets. We can better evaluate the generalization capability in different algorithms. We follow the experiment setup in Yue et al. (2020) and use the WRN2-28-10 as the backbone.

The backbone model is trained on all training classes using $\mathcal{C}$-class cross-entropy loss by the SGD optimizer (momentum of 0.9 and weight decay of 1e-4) with a mini-batch size of 64. The learning rate is initialized as 0.05 and is decayed by 0.1 after 60, 80, and 90 epochs (100 epochs in total). After the SGD training converges, we run 100 epochs of SWA with a learning rate of 0.02. Note that MFRL is not sensitive to training epochs and learning rates in SWA (see Appendix A.4). The training images are augmented with random crop, random horizontal flip, and color jitter.

During testing, we conduct 5 independent runs of 600 randomly sampled few-shot classification tasks from test classes and calculate the average accuracy. Each task contains 5 classes, $1 \times 5$ or $5 \times 5$ support samples, and 75 query samples. A logistic regression model is learned using only the support samples. The classification accuracy is evaluated on the query samples.

## A.3 ADDITIONAL RESULTS ON FEW-SHOT REGRESSION AND CLASSIFICATION

The additional results on few-shot regression using different activation functions are reported in Table 6. MFRL achieves high accuracy with different activation functions.

Table 6: 10-shot regression on sine waves with different activation functions.

| Sine wave | MSE |
|---|---|
| MFRL (ReLU activation) | $0.16 \pm 0.51$ |
| MFRL (tanh activation) | $0.018 \pm 0.011$ |
| MFRL (erf activation) | $0.016 \pm 0.008$ |

The few-shot classification results using a 4-layer convolutional neural network (or similar architectures) are reported in Table 7 and 8. Similar to the results using ResNet-12 and WRN-28-10, the proposed method outperforms a wide range of meta-learning approaches. Our method is only

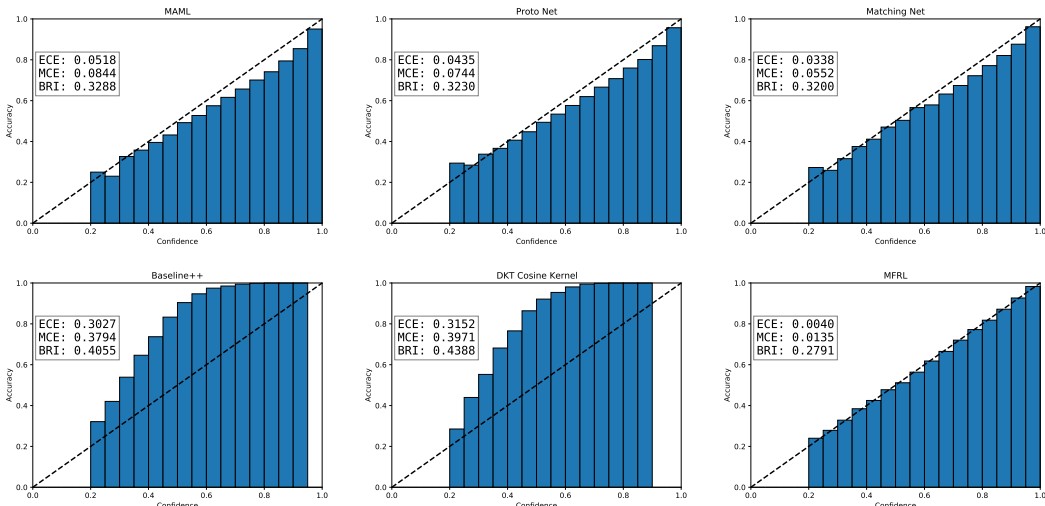

Figure 4: Reliability diagrams for 5-way 5-shot classification on miniImageNet

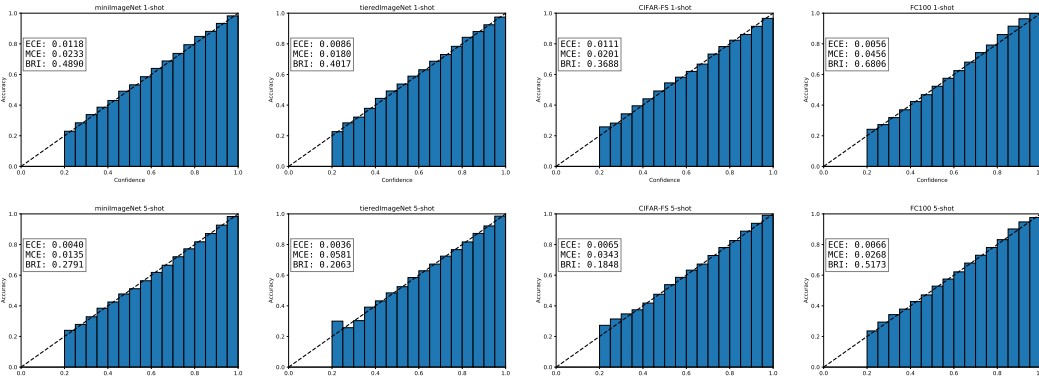

Figure 5: Reliability diagrams for 5-way few-shot classification using ResNet-12 backbone

Table 7: Few-shot classification results on miniImageNet and tieredImageNet.

| Method | Backbone | miniImageNet 5-way | | tieredImageNet 5-way | |
|---|---|---|---|---|---|
| | | 1-shot | 5-shot | 1-shot | 5-shot |
| Matching Net (Vinyals et al., 2016) | Conv-4 | $43.56 \pm 0.84$ | $55.31 \pm 0.73$ | $54.48 \pm 0.93$ | $71.32 \pm 0.78$ |
| Proto Net (Snell et al., 2017) | Conv-4 | $49.42 \pm 0.78$ | $68.20 \pm 0.66$ | $53.31 \pm 0.89$ | $72.69 \pm 0.74$ |
| MAML (Finn et al., 2017) | Conv-4 | $48.70 \pm 1.75$ | $63.11 \pm 0.92$ | - | - |
| SNAIL (Mishra et al., 2018) | Conv-4 | $45.10 \pm NA$ | $55.20 \pm NA$ | - | - |
| VERSA (Gordon et al., 2019) | Conv-5 | $53.40 \pm 1.82$ | $67.37 \pm 0.86$ | - | - |
| Meta Mixture (Jerfel et al., 2019) | Conv-4 | $49.60 \pm 1.50$ | $64.60 \pm 0.92$ | - | - |
| RelationNet (Sung et al., 2018) | Conv-4 | $50.44 \pm 0.82$ | $65.32 \pm 0.70$ | $54.48 \pm 0.93$ | $71.32 \pm 0.78$ |
| FPA (Qiao et al., 2018) | Conv-4 | $54.53 \pm 0.40$ | $67.87 \pm 0.20$ | - | - |
| Shot-free (Ravichandran et al., 2019) | Conv-4 | $49.07 \pm 0.43$ | $65.73 \pm 0.36$ | $48.19 \pm 0.43$ | $65.50 \pm 0.39$ |
| Baseline++ (Chen et al., 2019) | Conv-4 | $47.15 \pm 0.49$ | $66.18 \pm 0.18$ | $54.67 \pm 0.61$ | $72.37 \pm 0.67$ |
| FEAT (Ye et al., 2020) | Conv-4 | $\mathbf{55.15 \pm 0.20}$ | $\mathbf{71.61 \pm 0.16}$ | - | - |
| DKT (Patacchiola et al., 2020) | Conv-4 | $49.73 \pm 0.07$ | $64.00 \pm 0.09$ | - | - |
| MFRL | Conv-4 | $53.62 \pm 0.71$ | $71.52 \pm 0.60$ | $\mathbf{56.24 \pm 0.84}$ | $\mathbf{72.88 \pm 0.76}$ |

Table 8: Few-shot classification results on CIFAR-FS and FC100.

| Method | Backbone | CIFAR-FS 5-way | | FC100 5-way | |
|---|---|---|---|---|---|
| | | 1-shot | 5-shot | 1-shot | 5-shot |
| Proto Net (Snell et al., 2017) | Conv-4 | $55.5 \pm 0.7$ | $72.0 \pm 0.6$ | $35.3 \pm 0.6$ | $48.6 \pm 0.6$ |
| MAML (Finn et al., 2017) | Conv-4 | $58.9 \pm 1.9$ | $71.5 \pm 1.0$ | $38.1 \pm 1.7$ | $50.4 \pm 1.0$ |
| RelationNet (Sung et al., 2018) | Conv-4 | $55.0 \pm 1.0$ | $69.3 \pm 0.8$ | - | - |
| Shot-free (Ravichandran et al., 2019) | Conv-4 | $55.1 \pm 0.5$ | $71.7 \pm 0.4$ | - | - |
| Baseline++ (Chen et al., 2019) | Conv-4 | $55.1 \pm 0.9$ | $72.3 \pm 0.8$ | $35.2 \pm 0.7$ | $49.8 \pm 0.7$ |
| MFRL | Conv-4 | $\mathbf{64.3 \pm 0.9}$ | $\mathbf{79.4 \pm 0.5}$ | $\mathbf{40.1 \pm 0.8}$ | $\mathbf{54.4 \pm 0.7}$ |

second to few-shot embedding adaptation with transformer (FEAT) (Ye et al., 2020) on miniImageNet dataset. Recently, meta-learned attention modules are built on top of the convolutional neural network to get improved few-shot classification accuracy. Direct comparison to those methods with attention modules (Ye et al., 2020; Fei et al., 2021; Zhang et al., 2021) may not be fair because recent studies show that transformer itself can achieve better results than convolutional neural networks in image classification (Dosovitskiy et al., 2021). It is difficult to determine whether the performance improvement is due to the meta-learning algorithm or the attention modules. To make a fair comparison, we add convolutional block attention modules (Woo et al., 2018) on top of ResNet12 features (before global average pooling). As shown in Fig. 9, MFRL with attention modules achieves comparable results with MELR and IEPT.

The uncertainty calibration results of MFRL with the temperature scaling factor are presented in Fig. 5. The prediction confidence aligns well with the prediction accuracy. It demonstrates that MFRL with the temperature scaling factor results in well calibrated models.

## A.4 SENSITIVITY OF MFRL

The performance of MFRL is not sensitive to learning rates in SWA. As shown in Fig. 6, the representation learned by SWA generalizes better than the one from standard SGD, as long as the learning rate in SWA is in a reasonable range. In addition, the prediction accuracy on meta-test tasks keeps stable even after running SWA for many epochs on the training data. Therefore, MFRL is not sensitive to training epochs. This desirable property makes the proposed method easy to use when solving few-shot learning problems in practice.

Table 9: Results of MFRL with attention modules

| Method | Backbone | miniImageNet 5-way | | tieredImageNet 5-way | |
|---|---|---|---|---|---|
| | | 1-shot | 5-shot | 1-shot | 5-shot |
| FEAT (Ye et al., 2020) | ResNet-12 | $66.78 \pm 0.20$ | $82.05 \pm 0.14$ | $70.80 \pm 0.23$ | $84.79 \pm 0.16$ |
| MELR (Fei et al., 2021) | ResNet-12 | $67.40 \pm 0.43$ | $83.40 \pm 0.28$ | $72.14 \pm 0.51$ | $87.01 \pm 0.35$ |
| IEPT (Zhang et al., 2021) | ResNet-12 | $67.05 \pm 0.44$ | $82.90 \pm 0.30$ | $72.24 \pm 0.50$ | $86.73 \pm 0.34$ |
| MFRL | ResNet-12 | $67.18 \pm 0.79$ | $83.81 \pm 0.53$ | $71.58 \pm 0.79$ | $86.87 \pm 0.62$ |
| MFRL + Attention | ResNet-12 | $67.51 \pm 0.78$ | $83.97 \pm 0.51$ | $71.97 \pm 0.80$ | $86.99 \pm 0.60$ |

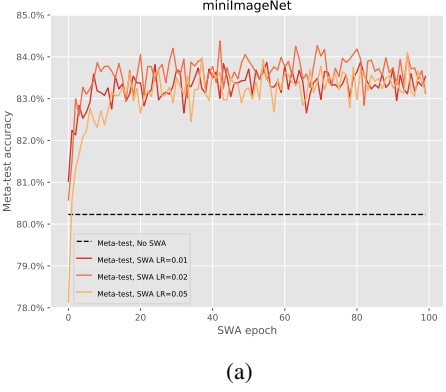
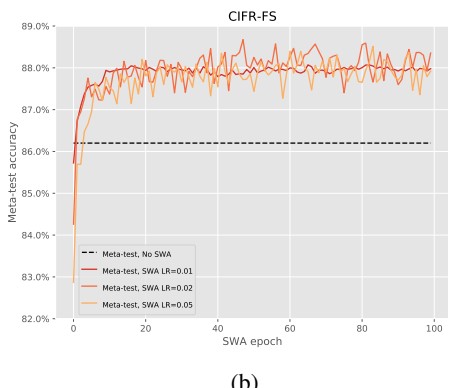

(a)  (b)

Figure 6: Evaluation on different learning rates and training epochs in SWA. (a) 5-way 5-shot accuracy on miniImageNet with ResNet12 backbone; (b) 5-way 5-shot accuracy on CIFAR-FS with WRN-28-10 backbone.

### A.5 COMPARISON WITH EXPONENTIAL MOVING AVERAGING

Exponential moving average (EMA) decays the importance of model weights from early training epochs exponentially. Let $\theta_{\text{avg}} \leftarrow a\theta_{\text{avg}} + (1-a)\theta_{\text{new}}$. We try EMA with different values of $a$. In Table 10, EMA improves the performance when $a$ is within a reasonable range. Note that EMA introduces one extra hyperparameter, the forgetting factor. It makes EMA less desirable in practice.

Table 10: Comparison between SWA and EMA

| Method | Backbone | miniImageNet 5-way | |
|---|---|---|---|
| | | 1-shot | 5-shot |
| No averaging | ResNet-12 | $62.27 \pm 0.86$ | $80.23 \pm 0.57$ |
| EMA $a = 0.9$ | ResNet-12 | $66.03 \pm 0.83$ | $82.90 \pm 0.58$ |
| EMA $a = 0.99$ | ResNet-12 | $66.05 \pm 0.84$ | $82.98 \pm 0.57$ |
| EMA $a = 0.999$ | ResNet-12 | $62.49 \pm 0.81$ | $80.73 \pm 0.62$ |
| SWA | ResNet-12 | $\mathbf{67.18 \pm 0.79}$ | $\mathbf{83.81 \pm 0.53}$ |

### A.6 HIERARCHICAL BAYESIAN LINEAR CLASSIFICATION MODEL

Similar to the hierarchical Bayesian linear regression model, the prior distribution over $\mathbf{w}$ is $p(\mathbf{w} \mid \lambda) = \prod_{i=0}^{p} \mathcal{N}(w_i \mid 0, \lambda)$, where $\lambda$ is the precision in the Gaussian prior. The hyperprior on $\lambda$ is defined as $p(\lambda) = \text{Gamma}(\lambda \mid a, b)$. The posterior over all latent variables given the data is

$$p(\mathbf{w}, \lambda \mid \mathbf{X}, \mathbf{y}) = \frac{p(\mathbf{y} \mid \mathbf{X}, \mathbf{w}, \lambda) \, p(\mathbf{w} \mid \lambda) \, p(\lambda)}{p(\mathbf{y} \mid \mathbf{X})} \tag{7}$$

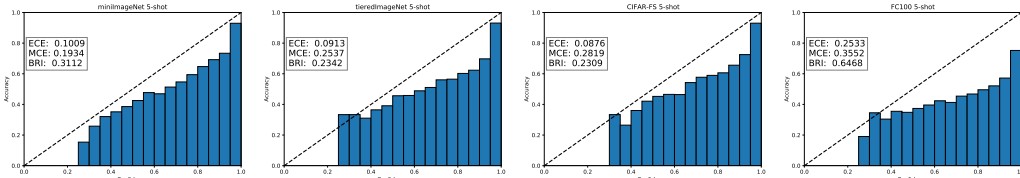

Figure 7: Reliability diagrams of hierarchical Bayesian linear classification models with flat and non-informative hyperpriors. The backbone is ResNet-12.

MCMC sampling (Hoffman & Gelman, 2014) is used to avoid potential deterioration in predictive performance due to approximated inference. A flat and non-informative hyperprior ($a = b = 10^{-6}$) is used because no prior knowledge is available. In Table 11, the hierarchical Bayesian linear classification model achieves slightly worse performance than the logistic regression model. However, the classification model is not well calibrated, as shown in Fig. 7.

Table 11: Comparison between logistic regression and hierarchical Bayesian linear models on 5-way 5-shot classification benchmarks using ResNet-12 backbone.

| Top layer | miniImageNet | tieredImageNet | CIFAR-FS | FC-100 |
|---|---|---|---|---|
| Logistic regression | 83.81 | 86.87 | 87.4 | 61.1 |
| Hierarchical Bayesian linear classification | 81.94 | 85.32 | 85.9 | 59.2 |

After fine-tuning $a$ and $b$ using the meta-validation data, it is possible to get better calibrated classification models on test tasks. Besides, the classification accuracy is still slightly worse than logistic regression after hyperparameter tuning. Our observations align with a recent study, which shows that the Bayesian classification model cannot achieve similar performance to the non-Bayesian counterpart without tempering the posterior (Wenzel et al., 2020). We do not further experiment tempered posterior in the hierarchical Bayesian linear classification model because it introduces an extra temperature hyperparameter that requires tuning. The original purpose of introducing the hierarchical Bayesian model is to get an accurate and well calibrated classification model without hyperparameter tuning. Consequently, the hierarchical Bayesian model is not used in few-shot classification in that hierarchical Bayesian linear classification models cannot achieve high accuracy and good uncertainty calibration from a non-informative hyperprior. If hyperparameter tuning is inevitable, it is much easier to tune a logistic regression model with a temperature scaling factor, compared with tuning a hierarchical Bayesian model. Furthermore, the computational cost of learning a hierarchical Bayesian linear classification model via MCMC sampling is much larger than that of learning a logistic regression model.

