# OpenReview forum: "Meta-free few-shot learning via representation learning with weight averaging"
_ICLR.cc/2022/Conference — ICLR 2022 Submitted_

### Official Review · Reviewer_7iTd · 2021-10-25

**Correctness:** 4
**Technical Novelty And Significance:** 3
**Empirical Novelty And Significance:** 3
**Recommendation:** 6
**Confidence:** 4

**Main Review:**

The writing is easy to follow. The introduction provides a good motivation for the proposed approach and presents a thorough review of the existing literature. The idea of using SWA for few-shot classification is itself interesting and new as far as I can tell, and the empirical results look promising. The conjecture that it fosters lower-rank representations is supported by empirical observations.

However, after reading the paper, I'm left with the question: beyond the application of SWA to the few-shot learning setting, what is MFRL? One of the claimed contributions is that it "can handle both few-shot regression and classification", however different adaptation strategies are used for regression (hierarchical Bayesian linear model) and classification (L2-regularized logistic regression with a temperature coefficient). As far as I can tell, the aspects common to both settings are i) learning the representation on the training data using a single objective rather than episodic losses (which has already been explored in previous work), and ii) applying SWA. In fact, Algorithm 1 feels more like two algorithms combined with if-statements than a unique algorithm. I feel that the presentation would have been better and more straightforwardly characterized as an application of SWA to the few-shot learning setting, and I'd be interested in hearing the authors' opinion on this.

Additional questions/comments:

- Can the authors say a few words on hyperparameter selection using meta-validation data? In the cross-domain setting (e.g. mini-ImageNet -> CUB, or Meta-Dataset), is this a good selection strategy?
- I am surprised by how well MFRL without SWA performs for regression. Can the authors expand on what aspect of MFRL is responsible for its good performance? As I understand, it consists in training concurrently on all training regression tasks; how is it that it outperforms more sophisticated meta-learning approaches?
- In the classification setting, MFRL only really stands out when SWA is applied. Are there other aspects to MFRL that positively impact performance?
- The difference between MFRL and Baseline++ (one normalizes the features, and the other normalizes both the features and the classification weights) is easy to miss. Being such a small implementation difference, it also reinforces the idea that MFRL reduces to SWA, and naming it is not necessary.
- Presenting results on model calibration is good, but the paper feels lacking in details. Judging by the introduction, using a temperature parameter is sufficient to obtain good model calibration. How come? The submission mentions that "given a meta-test task, a new  top layer is fine-tuned with few-shot samples to obtain probabilistic models with well-calibrated uncertainty"; what about the fine-tuning procedure ensures calibration? Is some kind of early-stopping performed by looking at calibration metrics?

**Summary Of The Paper:**

The submission introduces a few-shot classification and regression approach called Meta-Free Representation Learning (MFRL). First, a representation is learned on the meta-training data: for regression, the model is trained on all available training regression tasks concurrently; for classification, the model is trained on the full-ways classification problem using all training classes. Then, stochastic weight averaging (SWA) is applied to the model by continuing training for a certain number of epochs and averaging the parameters obtained across those additional epochs.

For test tasks, the regression or classification layer is discarded, and a new output layer is trained while freezing the network weights. For regression, approximate inference is performed on a hierarchical Bayesian linear model. For classification, a logistic regression model is trained with L2 regularization and a temperature hyperparameter.

Experimental results are presented on the sine wave and head pose problems for regression, and on mini-ImageNet, tiered-ImageNet, CIFAR-FS, and FC100 for classification. The extent to which SWA encourages learning lower-rank representations (as hypothesized) is verified through visualizations of the normalized eigenvalues. Finally, calibration curves are shown for classification to demonstrate how MFRL's temperature scaling factor leads to better calibrated models.

**Summary Of The Review:**

The submission applies an existing idea (SWA) to the few-shot learning setting, which is interesting and shows promising results. However, there is a disconnect between the simplicity of the idea and how it's presented by the submission, and overall there aren't a whole lot of new insights to be gained beyond the observation that SWA works well for few-shot learning.

---

**POST-REBUTTAL**: The response addresses most of my concerns. I would like to see those clarifications integrated in the main text.

---

> ### Author Response · Authors · 2021-11-16
> **Response to Reviewer 7iTd [1]**
>
> We appreciate the constructive comments from the reviewer. Below is our point-to-point response.
>
> *"However, after reading the paper, I'm left with the question: beyond the application of SWA to the few-shot learning setting, what is MFRL? One of the claimed contributions is that it "can handle both few-shot regression and classification", however different adaptation strategies are used for regression (hierarchical Bayesian linear model) and classification (L2-regularized logistic regression with a temperature coefficient). As far as I can tell, the aspects common to both settings are i) learning the representation on the training data using a single objective rather than episodic losses (which has already been explored in previous work), and ii) applying SWA. In fact, Algorithm 1 feels more like two algorithms combined with if-statements than a unique algorithm. I feel that the presentation would have been better and more straightforwardly characterized as an application of SWA to the few-shot learning setting, and I'd be interested in hearing the authors' opinion on this."*
>
> We use different losses and top layers due to the difference in the nature of classification and regression. Both hierarchical Bayesian linear model and L2-regularized logistic regression model are just linear models, with the difference in how the output variable is distributed (continuous Gaussian vs categorical). We package it in a single algorithm to emphasize that our algorithm can handle both regression and classification. Existing transfer learning based few-shot learning methods cannot handle few-shot regression. For example, cosine classifier in Baseline++ [1] and knowledge distillation[2] cannot be extended to handle regression problems.
>
> *"Can the authors say a few words on hyperparameter selection using meta-validation data? In the cross-domain setting (e.g. mini-ImageNet -> CUB, or Meta-Dataset), is this a good selection strategy?"*
>
> The learning rate and training epochs in SWA stage are two important hyperparameters. As shown in Fig 7 of the submitted paper, we find that the performance is robust as long as the learning rate in SWA stage is within a reasonable range.
> The temperature scaling factor is an important hyperparameter that affects the uncertainty calibration. We did grid search of scaling factor on the meta-validation set, and chose the temperature scaling factor resulting in best expected calibration error (ECE). We then applied the selected temperature scaling factor to the meta-test set.
> Here we show the results on the popular miniImageNet -> CUB cross-domain few-shot learning task, with comparison to other meta-learning methods. We follow the same experiment settings in [3] and use the same backbone WRN-28-10. We just use the same hyperparameter (training epochs, learning rate, learning rate in SWA, SWA epochs…) in miniImageNet dataset for miniImageNet -> CUB cross-domain dataset. Our method still achieves the best performance. It means that our method is very robust to hyperparameter choice.
>
> **miniImageNet->CUB 5-way classification**
>
> Method | 1-shot | 5-shot
> --- | --- | ---
> Baseline            | $42.89\pm0.41$ | $62.12\pm0.40$
> Baseline++          | $42.12\pm0.39$ | $60.21\pm0.39$
> MAML                | $39.06\pm0.47$ | $55.04\pm0.42$
> LEO                 | $41.45\pm0.54$ | $56.66\pm0.48$
> MTL                 | $43.15\pm0.44$ | $56.89\pm0.41$
> Matching Net        | $42.04\pm0.57$ | $53.08\pm0.45$
> SIB                 | $43.27\pm0.44$ | $59.94\pm0.42$
> OUrs                | $46.98\pm0.51$ | $66.92\pm0.42$
>
>
> *"I am surprised by how well MFRL without SWA performs for regression. Can the authors expand on what aspect of MFRL is responsible for its good performance? As I understand, it consists in training concurrently on all training regression tasks; how is it that it outperforms more sophisticated meta-learning approaches?"*
>
> We find that commonly used Relu activation function does not work well with few-shot regression. We replaced it with smooth activation function, such as tanh, and saw improvement as shown in Table 6 in the Appendix. Compared with few-shot classification, few-shot regression receives less attention. To our knowledge, none of the published papers compares against the performance of training concurrently on all training regression tasks. We believe our experiments show that training concurrently on all training regression tasks is a simple but strong baseline in few-shot regression.

---

> > ### Author Response · Authors · 2021-11-16
> > **Response to Reviewer 7iTd [2]**
> >
> > *"In the classification setting, MFRL only really stands out when SWA is applied. Are there other aspects to MFRL that positively impact performance?"*
> >
> > Everything is the same except for SWA. We find that SWA can reduce the rank of the representation. We believe the low-rank representation contributes to good performance.
> >
> > *"Presenting results on model calibration is good, but the paper feels lacking in details. Judging by the introduction, using a temperature parameter is sufficient to obtain good model calibration. How come? The submission mentions that "given a meta-test task, a new top layer is fine-tuned with few-shot samples to obtain probabilistic models with well-calibrated uncertainty"; what about the fine-tuning procedure ensures calibration? Is some kind of early-stopping performed by looking at calibration metrics?"*
> >
> > We have to tune two hyperparameters in logistic regression: the L2 regularization coefficient and the temperature scaling factor. At first, we set the temperature scaling factor to 1, and do grid search on the meta-validation to find the L2 regularization coefficient resulting in the highest meta-validation accuracy. However, fine-tuning an L2 regularization coefficient does not ensure good calibration. It is the temperature scaling factor that ensures good uncertainty calibration. Similarly, we do grid search of temperature scaling factor on the meta-validation set, and choose the temperature scaling factor resulting in the lowest expected calibration error (ECE). Then we just apply the selected temperature scaling factor to the meta-test set. Note that the temperature scaling factor does not affect the accuracy in classification.
> >
> > [1] Chen, Wei-Yu, et al. "A Closer Look at Few-shot Classification." International Conference on Learning Representations. 2019.
> >
> > [2] Tian, Yonglong, et al. "Rethinking few-shot image classification: a good embedding is all you need?." Computer Vision–ECCV 2020: 16th European Conference, Glasgow, UK, August 23–28, 2020.
> >
> > [3] Yue, Zhongqi, et al. "Interventional Few-Shot Learning." Advances in Neural Information Processing Systems 33 (2020).

---

> > > ### Comment · Reviewer_7iTd · 2021-11-19
> > > **Updated review**
> > >
> > > Thank you for your response. It addresses most of my concerns, and I updated my score to reflect that. However I would like to see those clarifications integrated in the main text before I consider raising my score further.
> > >
> > > RE: using the name MFRA to describe SWA being applied to few-shot learning. Without disqualifying the paper, I think this could cause confusion for readers who (like me) assume that there must be a key difference between MFRA and SWA if they have different names.

---

> > > > ### Author Response · Authors · 2021-11-21
> > > > **We upload the revised paper**
> > > >
> > > > We appreciate the reply from the reviewer. We upload the revised paper. We rewrite part of the introduction to clarify that the proposed method is just a transfer learning method and try to avoid any confusion. We keep the name MFRL for now because we are not able to figure out an appropriate name in a short period of time. We seriously consider a more appropriate name to avoid any possible confusion in the next draft.

---

### Official Review · Reviewer_useT · 2021-11-03

**Correctness:** 3
**Technical Novelty And Significance:** 2
**Empirical Novelty And Significance:** Not applicable
**Recommendation:** 5
**Confidence:** 3

**Main Review:**

This paper presents a few-shot learning method based on the reprehensive pre-train learning using stochastic weight averaging (SWA). The merits of the proposed work is that this proposed method can works for both few-shot regression and classification problems and both of them achieves better results compared with the other recent works as reported in the manuscript. And as claimed by the author, this is the first few-shot learning work can works for classification and regression.

1), It is well-known that SWA is an effective tool in weight space to obtain better reprehensive learning as it leads to a more flat minimum. As a result, it is not very supervising that SWA achieves better results in transfer learning for few-learning learning. Actually, SWA improves most applications as the author mentioned for meta-based approaches. Therefore, employing SWA for few-shot learning might not be a contribution that significant enough for an individual work. It will be more interesting to provide more analyse why SWA lead to the better performance compared with other solutions. Moreover, the flat loss surface contribute to the adaption is well discussed in some existing incremental few-shot learning research.

2), It is interesting that the authors use the theory for meta learning to guide the research for transfer learning based few-shot learning, i.e., to make the feature with lower-rank representation. The result shows that the performance is improved. However, the theoretical link is still not clear. Is the low-rank representation criterion general for all reprehensive learning to have better result? Or the SWA is best way to achieve the low-rank representation for few-shot learning? It might be suspected that all those learning methods resulting better performance will contribute to the low-rank representation, e.g., various pre-training losses, network structure, etc.

3), For the transfer learning based few-shot learning method, there will be a problem that by freezing the feature extraction, it is almost impossible to further adapt and generalize to cross-domain test environment if the feature difference between base classes and few-shot classes are significant. As a result, the evaluation performance will be very related to the dataset whether the few-shot/pretrain data are similar or not. In contrast, methods like MAML will adapt and finetune the feature extraction for the novel classes, which leads it may more difficult to train. However, its generalization capability will be stronger when the feature of few-shot classes are different with the pretrain base classes. As a result, to prove the effectiveness of the proposed methods, more ablation should be added with different setting, e.g., the few-shot classes close to the pretrain base classes, few-shot classes far away to the pretrain base classes, etc. In this way, the reader will be able to see the comparative advantages and limitations of the proposed method. Otherwise, it might not be proper to directly conclude the proposed methods is better than the existing methods.

4), Similar like SWA, Exponential Moving Average (EMA) has the similar better results than the directly trained weights, and both are approaches in the weight space. It will be reasonable and interesting to be an ablation study to see if EMA achieves the similar results in few-shot learning or if EMA also contributes to the low-rank representation.

5), Similarly, for other transfer learning based few-shot learning method such as Baseline++, it is straightforward and fully compatible to employ the SWA in its pertrain, the comparison should be analysed if MFRL better than (Baseline++ with SWA) to demonstrate the effectiveness of the MFRL.

**Summary Of The Paper:**

This paper presents a few-shot learning method based on the reprehensive pre-train learning using stochastic weight averaging (SWA). The merits of the proposed work is that this proposed method can works for both few-shot regression and classification problems and both of them achieves better results compared with the other recent works as reported in the manuscript. And as claimed by the author, this is the first few-shot learning work can works for classification and regression.


**Summary Of The Review:**

As mentioned in the Main Review above, the merits and concerns are pointed for reference and discussion. Generally, the main concern is on the originality as the SWA is an existing method, and the detail analysis on the mechanism how SWA specifically contribute for few-shot learning is not adequate.

---

> ### Author Response · Authors · 2021-11-16
> **Response to Reviewer useT [1]**
>
> We appreciate the reviewer for the constructive comments. Below is our point-to-point response.
>
> *"It is well-known that SWA is an effective tool in weight space to obtain better reprehensive learning as it leads to a more flat minimum. As a result, it is not very supervising that SWA achieves better results in transfer learning for few-learning learning. Actually, SWA improves most applications as the author mentioned for meta-based approaches. Therefore, employing SWA for few-shot learning might not be a contribution that significant enough for an individual work. It will be more interesting to provide more analyse why SWA lead to the better performance compared with other solutions. Moreover, the flat loss surface contribute to the adaption is well discussed in some existing incremental few-shot learning research."*
>
> In Section 6, we discuss that convergence to flat minimum would not explain the effectiveness of SWA in few-shot learning. This is because the meta-training and meta-testing losses are not comparable once the top layer is re-trained by the few-shot support data in a meta-test task. Therefore, the effectiveness of SWA in few-shot learning is more likely related to the property of the representation. Although our results empirically demonstrate that SWA results in low-rank representation, further research about the connection is needed.
>
> To the best of our knowledge, the low-rank bias of SWA is not mentioned in any previous papers.
>
> *"It is interesting that the authors use the theory for meta learning to guide the research for transfer learning based few-shot learning, i.e., to make the feature with lower-rank representation. The result shows that the performance is improved. However, the theoretical link is still not clear. Is the low-rank representation criterion general for all reprehensive learning to have better result? Or the SWA is best way to achieve the low-rank representation for few-shot learning? It might be suspected that all those learning methods resulting better performance will contribute to the low-rank representation, e.g., various pre-training losses, network structure, etc."*
>
> Whether low-rank representation is a general criterion for good representation learning is an open question. Some papers empirically show that low-rank representation is helpful in generative models [1] and supervised learning [2]. In Section 6, we discussed a few alternatives to achieve low-rank representation, such as L1 regularization, nuclear norm, spectral norm, and Frobenius norm regularization. We choose SWA because it is compatible with standard SGD training in supervised learning and meta-learning. Most low-rank regularization methods are explicit, but it is difficult to choose the appropriate strength of explicit regularization. If too strong, explicit regularization can bias towards simple solutions that do not fit the data. SWA is an implicit regularization towards low-rank representation so that we do not have to worry about hyperparameter tuning.
>
> We do not know if SWA is the best way to achieve low-rank representation. It is an interesting topic to examine the effectiveness of SWA and alternative low-rank regularization.  We agree with the reviewer that various losses and network structure in few-shot learning may just result in low-rank representation. To validate this hypothesis, it requires re-implementation of a wide range of existing few-shot learning methods under the same experiment environment and comparing the rank of representation. It is an interesting topic that helps us understand important open questions in this research area, but it is beyond the scope of this paper.

---

> > ### Author Response · Authors · 2021-11-16
> > **Response to Reviewer useT [2]**
> >
> > *"For the transfer learning based few-shot learning method, there will be a problem that by freezing the feature extraction, it is almost impossible to further adapt and generalize to cross-domain test environment if the feature difference between base classes and few-shot classes are significant. As a result, the evaluation performance will be very related to the dataset whether the few-shot/pretrain data are similar or not. In contrast, methods like MAML will adapt and finetune the feature extraction for the novel classes, which leads it may more difficult to train. However, its generalization capability will be stronger when the feature of few-shot classes are different with the pretrain base classes. As a result, to prove the effectiveness of the proposed methods, more ablation should be added with different setting, e.g., the few-shot classes close to the pretrain base classes, few-shot classes far away to the pretrain base classes, etc. In this way, the reader will be able to see the comparative advantages and limitations of the proposed method. Otherwise, it might not be proper to directly conclude the proposed methods is better than the existing methods."*
> >
> > We appreciate the reviewer for bringing the discussion about the limitations of transfer learning and potential advantages of meta-learning. Ideally, meta-learning methods with adaptive embedding should have strong generalization capability when the base classes and novel classes are quite different. Meta-dataset [5], a very large-scale dataset for cross-domain few-shot learning from Google Research, could be a good testbed to check the performance of the model when the similarity between base and novel classes varies. Surprisingly, simple transfer learning methods still outperform meta-learning methods with adaptive embedding according to Table 7 in [3]. However, we do not have enough computation resource to finish the experiment on Meta-dataset within a short period of time. Here we show the results on the popular miniImageNet -> CUB cross-domain few-shot learning task, with comparison to some meta-learning methods. We follow the same experiment settings in [4] and use the same backbone WRN-28-10. We can see that Baseline and Baseline++ already outperform those meta-learning methods. Our method achieves the best performance.
> >
> > We believe that a model with adaptive capability is helpful when the base and novel classes are very different. However, current episodic meta-training scheme does not result in satisfactory results. How to properly train a model with adaptive capability is still an open question.
> >
> >
> > **miniImageNet->CUB 5-way classification**
> >
> > Method | 1-shot | 5-shot
> > --- | --- | ---
> > Baseline            | $42.89\pm0.41$ | $62.12\pm0.40$
> > Baseline++          | $42.12\pm0.39$ | $60.21\pm0.39$
> > MAML                | $39.06\pm0.47$ | $55.04\pm0.42$
> > LEO                 | $41.45\pm0.54$ | $56.66\pm0.48$
> > MTL                 | $43.15\pm0.44$ | $56.89\pm0.41$
> > Matching Net        | $42.04\pm0.57$ | $53.08\pm0.45$
> > SIB                 | $43.27\pm0.44$ | $59.94\pm0.42$
> > OUrs                | $46.98\pm0.51$ | $66.92\pm0.42$

---

> > > ### Author Response · Authors · 2021-11-16
> > > **Response to Reviewer useT [3]**
> > >
> > > *"Similar like SWA, Exponential Moving Average (EMA) has the similar better results than the directly trained weights, and both are approaches in the weight space. It will be reasonable and interesting to be an ablation study to see if EMA achieves the similar results in few-shot learning or if EMA also contributes to the low-rank representation."*
> > >
> > > Here we show the results using EMA and comparison with SWA. Let $\theta_{avg}\gets a\theta_{avg}+\left(1-a\right)\theta_{new}$.  We try EMA with different values of $a$. We find that EMA improves the performance and results in low-rank representation when $a$ is within a reasonable range. Note that EMA introduces one extra hyperparameter, the forgetting factor. It makes EMA less desirable in practice.
> > >
> > > **miniImageNet 5-way classification**
> > >
> > > Method | 1-shot | 5-shot
> > > --- | --- | ---
> > > No averaging        | $62.27\pm0.86$ | $80.23\pm0.57$
> > > EMA a=0.9           | $66.03\pm0.83$ | $82.90\pm0.58$
> > > EMA a=0.99          | $66.05\pm0.84$ | $82.98\pm0.57$
> > > EMA a=0.99          | $62.49\pm0.81$ | $80.73\pm0.62$
> > > SWA                 | $67.18\pm0.79$ | $83.81\pm0.53$
> > >
> > > *"Similarly, for other transfer learning based few-shot learning method such as Baseline++, it is straightforward and fully compatible to employ the SWA in its pretrain, the comparison should be analysed if MFRL better than (Baseline++ with SWA) to demonstrate the effectiveness of the MFRL."*
> > >
> > > Below we show the results on Baseline++ with SWA. Note that Baseline++ normalizes both the weights and features on the top layer, while our method only normalizes the features. Our method performs slightly better.
> > >
> > > Method | miniImageNet | 5-way         | tieredImageNet  | 5-way
> > > --- | --- | --- | --- | ---
> > > $\,$            | 1-shot         | 5-shot         | 1-shot         | 5-shot
> > > Baseline++ SWA  | $65.72\pm0.80$ | $81.26\pm0.68$ | $70.01\pm0.82$ | $84.39\pm0.64$
> > > MFRL            | $67.18\pm0.79$ | $83.81\pm0.53$ | $71.58\pm0.79$ | $86.87\pm0.62$
> > >
> > > [1] Jing, Li, and Jure Zbontar. "Implicit Rank-Minimizing Autoencoder." Advances in Neural Information Processing Systems 33 (2020).
> > >
> > > [2] Huh, Minyoung, et al. "The Low-Rank Simplicity Bias in Deep Networks." arXiv preprint arXiv:2103.10427 (2021).
> > >
> > > [3] Tian, Yonglong, et al. "Rethinking few-shot image classification: a good embedding is all you need?." Computer Vision–ECCV 2020: 16th European Conference, Glasgow, UK, August 23–28, 2020.
> > >
> > > [4] Yue, Zhongqi, et al. "Interventional Few-Shot Learning." Advances in Neural Information Processing Systems 33 (2020).
> > >
> > > [5] Triantafillou, Eleni, et al. "Meta-Dataset: A Dataset of Datasets for Learning to Learn from Few Examples." International Conference on Learning Representations. 2020.

---

### Official Review · Reviewer_jVyb · 2021-11-04

**Correctness:** 3
**Technical Novelty And Significance:** 2
**Empirical Novelty And Significance:** 2
**Recommendation:** 5
**Confidence:** 4

**Main Review:**

There are several concerns that needs to be addressed, and I believe addressing these can improve the quality of this paper.




- There are some typos in the main text, like Prot Net in 5.5.




I have some major concerns about the opening remarks of this paper listed below:




- In the introduction, the authors have mentioned that : “Despite the success of episodic meta-learning in few-shot learning tasks, they are slow to converge, prone to over-fitting, and tricky to implement (Antoniou et al., 2018 [1]).”

I think this statement is not generally true, as these are the shortcoming of the MAML algorithm listed in [1], and most of them are addressed in this paper as MAML++. Also, as an example recent metric-based methods (ProtoNet, RelationNet, FT for cross-domain FSL [2]) does not suffer from these issues that much, at least when comparing to the transfer learning methods like Chen et al. [3] and Dhillon et al [4].

For example, ProtoNet, has a relatively fast convergence and increasing the ways in meta-training is proved to resolve overfitting. There are also several recent extensions, and reqularizers for few-shot learning that address these problems well.




- In the later part of the introduction, you also mentioned that your algorithm tries to address these shortcomings: “In this paper, we propose a new method that does not rely on meta-training to overcome the limitations of commonly used episodic meta-learning approaches in few-shot learning.”. But, there is no evaluation regarding faster convergence, preventing overfitting, and simpler implementation of your algorithm compared to recent episodic ones.




- I also doubt the validity of this statement in the introduction: “recent studies on transfer learning (Chen et al., 2019; Tian et al., 2020) cast doubt on whether it is the episodic meta-learning algorithm or the learned representation that is responsible for fast adaption to new tasks”.

Specifically, I am not so sure about differentiating episodic meta-learning and learning better features. As an example, Goldblum et al. [5] show that meta-learned features have a better generalization compared to conventionally trained networks with exactly the same structure. So, using the meta-learning algorithms leads to the better representation learning and these two are not to independent concepts.




- What do you exactly mean by this?

“Our method is the first one to achieve well-calibrated few-shot models by only fine-tuning probabilistic linear models in the meta-test phase, without any learning mechanisms related to the meta-training or representation learning phase.”

This mechanism is the well-known solution in transfer learning methods which is firstly popularized by [3] for few-shot learning. Am I missing something here?




Considering all these problems, I think the authors need to rewrite the introduction part and make it more reliable, considering the recent studies in few-shot learning. There are also some concerns for the Methodology part:




- Based on my understanding, your representation learning part is similar to [3], and the only difference is the usage of SWA as post-processinf of the network parameters. Although it is mentioned that the purpose is to find a low-rank approximation, it is not clear that how do you finally deploy it in your algorithm. The algorithm is unclear and seems to be lost in the explanation of various concepts!

Are you using a simple averaging (mentioned in equation (1)) as a regularizer and just describing that this implicitly steers the model towards low-rank representation?




- For the fine-tuning in few-shot classification, it is not clear for me how logistic regression model fits within your framework.




- I think adapting the temperature scaling for meta-test phase is the only contribution for fine-tuning few-shot classification. Is this new for few-shot learning to be considered as new algorithm?




- Overall, I found it very hard to follow the methodology as in some cases, the main flow of the algorithm is lost due to overexplanation of other concepts.




Regarding experiments:




- The few-shot regression results are interesting, but the few-shot classification results are not fair, because some recent algorithms like MELR [6] are not included in the comparison.







References:

[1] How to train your MAML, ICLR 2018.

[2] Cross-Domain Few-shot Classification via Learned Features-wise Transformation, ICLR 2020.

[3] A closer look at few-shot classification, ICLR 2019.

[4] A Baseline for Few-Shot Image Classification, ICLR 2020.

[5] Unraveling Meta-Learning: Understanding Feature Representations for Few-Shot Tasks, ICML 2020.

[6] MELR: Meta-Learning via Modeling Episode-Level Relationships for Few-shot Learning, ICLR 2021.




**Summary Of The Paper:**

This paper proposes a transfer learning method for few-shot regression/classification. In representation learning, it adapts the stochastic weight averaging (SWA) as a regularizer for learning more generalizable features. In fine-tuning phase (adaption to a new task in the meta-test phase), it treats the regression and classification differently. For regression a hierarchical Bayse is used while in classification, a conventional method is merged with temperature scaling. This work achieves comparable results to SOTA, however, they are major concerns that need to be addressed.

**Summary Of The Review:**

In summary, the proposed algorithm seems interesting. However, there are some major problems in the opening remarks of the paper, the explanation of the algorithm is not clear enough, and the few-shot classification performance of the proposed method is not fairly compared with recent methods. These concerns need to be addressed.

---

> ### Author Response · Authors · 2021-11-16
> **Response to Reviewer jVyb [1]**
>
> We appreciate the reviewer’s comments to improve the quality of the paper. Below is our point-to-point response.
>
> *"In the introduction, the authors have mentioned that: “Despite the success of episodic meta-learning in few-shot learning tasks, they are slow to converge, prone to over-fitting, and tricky to implement (Antoniou et al., 2018 [1])."I think this statement is not generally true, as these are the shortcoming of the MAML algorithm listed in [1], and most of them are addressed in this paper as MAML++. Also, as an example recent metric-based methods (ProtoNet, RelationNet, FT for cross-domain FSL [2]) does not suffer from these issues that much, at least when compared to the transfer learning methods like Chen et al. [3] and Dhillon et al [4]."*
>
> We appreciate the reviewer for raising a different view on meta-learning methods, which we beg to differ as follows: First, we do not believe MAML++ fully addresses the shortcomings of MAML since MAML++ heuristically switches from first-order derivatives to second-order derivatives as training progresses to save computational cost. It is known that computational cost of second-order derivatives is huge. Therefore, MAML-type algorithms with second-order derivatives will converge slower in time than methods without using second order derivatives. Moreover, MAML++ does not compare with strong baseline metric-based methods such as ProtoNet. In literature, most MAML-type algorithms do not report results on tired-ImgageNet with Reset12 backbone, which we believe is due to the prohibitively high computational cost.
> For metric-based methods, over-fitting on meta-training data still exists, despite that the literature remains relatively quiet on this. It is reported in the Figure 2 & 3 in Meta-baseline [1], as well as we observe similar over-fitting behavior of metric-based methods such as ProtoNet in our re-implementation. As shown in Figure 7 in our submitted paper, our method does not show any sign of over-fitting when training with extra hundreds of epochs.
> Finally, it is no doubt that meta-learning methods are more difficult to implement compared with transfer learning methods. It is easy to implement transfer learning methods like Baseline++ because we do not have to consider constructing episodic support data, episodic query data, inner loop, outer loop, etc, during training.
> As such, we believe that transfer learning methods are faster to converge, less prone to over-fitting, and easier to implement. Note that we are not advocating transfer learning as the best way for few-shot learning problems, but just want to explore whether it is possible to use simple methods without meta-learning to achieve comparable or better results of complicated meta-learning methods in few-shot learning.
>
> *"For example, ProtoNet, has a relatively fast convergence and increasing the ways in meta-training is proved to resolve overfitting."*
>
> The “higher-way” training in ProtoNet is an interesting topic. It is reported in the literature that training ProtoNet with higher-way results in better results. If we extend this idea to the extreme, the “highest way” we can use is the total number of classes in the meta-training dataset. In this extreme case, we are just doing the same thing as transfer learning.
>
> *"In the later part of the introduction, you also mentioned that your algorithm tries to address these shortcomings: “In this paper, we propose a new method that does not rely on meta-training to overcome the limitations of commonly used episodic meta-learning approaches in few-shot learning.”. But, there is no evaluation regarding faster convergence, preventing overfitting, and simpler implementation of your algorithm compared to recent episodic ones."*
>
> Our algorithm is in the category of transfer learning. It is much easier to implement transfer learning methods in practice, compared with meta-learning methods because the pretraining stage of transfer learning does not involve episodic support data, episodic query data, inner loop, outer loop… Figure 7 in our submitted paper shows that the proposed method is robust to over-fitting. Moreover, our algorithm does not include any second-order derivatives, resulting in faster convergence than methods using second-order derivatives.

---

> > ### Author Response · Authors · 2021-11-16
> > **Response to Reviewer jVyb [2]**
> >
> > *"I also doubt the validity of this statement in the introduction: “recent studies on transfer learning (Chen et al., 2019; Tian et al., 2020) cast doubt on whether it is the episodic meta-learning algorithm or the learned representation that is responsible for fast adaption to new tasks”. Specifically, I am not so sure about differentiating episodic meta-learning and learning better features. As an example, Goldblum et al. [5] show that meta-learned features have a better generalization compared to conventionally trained networks with exactly the same structure. So, using the meta-learning algorithms leads to the better representation learning and these two are not to independent concepts."*
> >
> > We appreciate that the reviewer brings to the discussion the link and difference between meta-learning and transfer learning. Transfer learning just learns a N-way classification model on the entire training dataset, where N is the total number of training classes. Meta-learning trains the model with episodic tasks. Meta-learning is proposed because it is assumed that transfer learning cannot well handle few-shot learning problems. We do not claim that episodic meta-learning cannot learn good representation. We just doubt whether it is necessary to use episodic meta-learning to obtain good features or representation. If we can obtain good features for a novel few-shot task without episodic meta-learning, we do not have to use episodic meta-learning.
> >
> > Goldblum et al. contradicts with the results in two popular transfer learning papers [2, 3], which show simple transfer learning can achieve comparable or better results than SOTA meta-learning methods. We reimplement the code in [2, 3] only to confirm the findings in the two papers. Based on the results from [2, 3] and results from our own algorithm, we know that transfer learning methods can achieve SOTA performance in few-shot learning without using meta-learning. Therefore, it is reasonable to question whether it is necessary to use meta-learning in few-shot learning given the extra efforts required.
> >
> > *"What do you exactly mean by this? “Our method is the first one to achieve well-calibrated few-shot models by only fine-tuning probabilistic linear models in the meta-test phase, without any learning mechanisms related to the meta-training or representation learning phase.” This mechanism is the well-known solution in transfer learning methods which is firstly popularized by [3] for few-shot learning. Am I missing something here?"*
> >
> > We address the uncertainty calibration problem for few-shot learning model, which is not addressed by Baseline++ at all. It is related to Figure 3, 5 & 6 in the submitted paper.
> >
> > *"Are you using a simple averaging (mentioned in equation (1)) as a regularizer and just describing that this implicitly steers the model towards low-rank representation?"*
> >
> > The pseudo code of our method is presented in Appendix A1. We just use a simple averaging of the model weights to steer the model towards the low-rank representation.
> >
> > *"For the fine-tuning in few-shot classification, it is not clear for me how logistic regression model fits within your framework."*
> >
> > Our algorithm is similar to Baseline++. In the meta-test phase, we freeze the parameters in the feature extractor. We just train a logistic regression between the encoded features and the targets using the few-shot support data.
> >
> > *"I think adapting the temperature scaling for meta-test phase is the only contribution for fine-tuning few-shot classification. Is this new for few-shot learning to be considered as new algorithm?"*
> >
> > Introducing a temperature scaling factor alone would not make a new algorithm. We use SWA to post-process the model weights to achieve stronger performance in few-shot learning tasks, compared with highly complicated meta-learning methods. Our method is the first transfer learning method that can handle both few-shot regression and classification. In few-shot regression, our method outperforms previous SOTA by a large margin. Considering the simplicity of the proposed method, it is very useful in solving practical few-shot learning problems. As a result, we do not agree that adapting the temperature scaling in few-shot classification is the only contribution of the paper.

---

> > > ### Author Response · Authors · 2021-11-16
> > > **Response to Reviewer jVyb [3]**
> > >
> > > *"The few-shot regression results are interesting, but the few-shot classification results are not fair, because some recent algorithms like MELR [6] are not included in the comparison."*
> > >
> > > It is not fair to directly compare the proposed method and MELR because MELR has an attention module on top the CNN features. To make a fair comparison, we add a convolutional block attention module [4] on top of ResNet12 features (before global average pooling). MFRL + Attention achieves comparable results with MERL. Note that MERL does not release their source code.
> > >
> > > Method | miniImageNet | 5-way         | tieredImageNet  | 5-way
> > > --- | --- | --- | --- | ---
> > > $\,$            | 1-shot         | 5-shot         | 1-shot         | 5-shot
> > > MELR            | $67.40\pm0.43$ | $83.40\pm0.28$ | $72.14\pm0.51$ | $87.01\pm0.35$
> > > MFRL            | $67.18\pm0.79$ | $83.81\pm0.53$ | $71.58\pm0.79$ | $86.87\pm0.62$
> > > NFRL + Attention| $67.51\pm0.78$ | $83.97\pm0.51$ | $71.97\pm0.80$ | $86.99\pm0.60$
> > >
> > >
> > > [1] Chen, Yinbo, et al. "Meta-Baseline: Exploring Simple Meta-Learning for Few-Shot Learning." Proceedings of the IEEE/CVF International Conference on Computer Vision. 2021.
> > >
> > > [2] Chen, Wei-Yu, et al. "A Closer Look at Few-shot Classification." International Conference on Learning Representations. 2019.
> > >
> > > [3] Tian, Yonglong, et al. "Rethinking few-shot image classification: a good embedding is all you need?." Computer Vision–ECCV 2020: 16th European Conference, Glasgow, UK, August 23–28, 2020, Proceedings, Part XIV 16. Springer International Publishing, 2020.
> > >
> > > [4] Woo, Sanghyun, et al. "Cbam: Convolutional block attention module." Proceedings of the European conference on computer vision (ECCV). 2018.

---

### Author Response · Authors · 2021-11-21
**The revised paper is uploaded**

We appreciate all reviewers for the insightful comments to improve the quality of the paper.

We upload the revised paper to address reviewers’ questions and concerns. We also made point-to-point responses to each reviewer. We rewrite part of the introduction to avoid confusion. We are happy to make further responses if necessary.

---

### Decision · Program_Chairs · 2022-01-20

**Decision:**

Reject

**Comment:**

By the scores, this submission is quite borderline. This paper introduces stochastic weight averaging into a few-shot learning setting, The reviewers all agreed the work was sound; discussion after the author response focused on the theoretical justifications, degree of novelty and potential impact, and the empirical support.

The primary concerns were that the work was slightly too incremental to obviously merit publication at this stage: though the empirical results were sound, they mostly follow the existing observation that SWA tends to be beneficial for generalization in other settings; apparently in few-shot learning as well. The positives would be that this is simple enough that it could become a general "best practice" in few-shot learning baselines, and as such communicating this is important.

The other discussion focused around the theoretical justification relating SWA to low-rank solutions. While empirically it does seem that the solutions found by SWA lead to low-rank representations, this is not really adequately explored, and it's not clear enough why this should be expected to happen. I think if this relationship between SWA and low-rank representations were more clearly explored then the paper would be a strong accept.

As it stands, it is quite borderline. Based on the scores (5,5,6), the recommendation is to reject, but it certainly could be included as well, as it has solid execution and is a clear topical fit for ICLR.